# Co-occupancy identifies transcription factor co-operation for axon growth

Ishwariya Venkatesh 🆔 [1,2✉], Vatsal Mehra[1], Zimei Wang[1], Matthew T. Simpson[1], Erik Eastwood[1], Advaita Chakraborty[1], Zac Beine[1], Derek Gross[1], Michael Cabahug[1], Greta Olson[1] & Murray G. Blackmore 🆔 [1,2✉]

Transcription factors (TFs) act as powerful levers to regulate neural physiology and can be targeted to improve cellular responses to injury or disease. Because TFs often depend on cooperative activity, a major challenge is to identify and deploy optimal sets. Here we developed a bioinformatics pipeline, centered on TF co-occupancy of regulatory DNA, and used it to predict factors that potentiate the effects of pro-regenerative Klf6 in vitro. High content screens of neurite outgrowth identified cooperative activity by 12 candidates, and systematic testing in a mouse model of corticospinal tract (CST) damage substantiated three novel instances of pairwise cooperation. Combined Klf6 and Nr5a2 drove the strongest growth, and transcriptional profiling of CST neurons identified Klf6/Nr5a2-responsive gene networks involved in macromolecule biosynthesis and DNA repair. These data identify TF combinations that promote enhanced CST growth, clarify the transcriptional correlates, and provide a bioinformatics approach to detect TF cooperation.

[1] Department of Biomedical Sciences, Marquette University, Milwaukee, WI, USA. [2]These authors contributed equally: Ishwariya Venkatesh, Murray G. Blackmore. ✉email: ishwariya.venkatesh@marquette.edu; murray.blackmore@marquette.edu

As they mature, neurons in the central nervous system (CNS) decline in their capacity for robust axon growth, which broadly limits recovery from injury[1–5]. Axon growth depends on the transcription of large networks of regeneration-associated genes (RAGs) and coaxing activation of these networks in adult CNS neurons is a major unmet goal[2,4]. During periods of developmental axon growth, RAG expression is supported by pro-growth transcription factors (TFs) that bind to relevant promoter and enhancer regions, to activate transcription[3,6]. One factor that limits RAG expression in mature neurons is the developmental downregulation of these pro-growth TFs[2,4]. Thus, identifying TFs that act developmentally to enable axon growth and supplying them to mature neurons is a promising approach to improve regenerative outcomes in the injured nervous system.

An ongoing challenge, however, is to decode the optimal set of factors for axon growth. To date, progress has centered on identifying individual TFs whose ectopic expression in mature neurons leads to improved axon growth. For example, we showed previously that the TF Klf6 is developmentally downregulated, and that viral re-expression drives improved axon growth in mature corticospinal neurons, which are critical mediators of fine movement[7]. The restoration of axon growth from this and other single-factor studies remains partial, however, indicating the need for additional intervention[7–12]. One likely possibility is that, because multiple TFs generally act in a coordinated manner to regulate transcription, a more complete restoration of axon growth will depend on multiple TFs. Consistent with this, regeneration-competent neurons recruit groups of interacting TFs in the hours to days post injury, which results in transcriptional remodeling, leading to reactivation of growth gene networks and functional recovery[13–15]. The need for multi-TF interventions in CNS neurons is recognized conceptually[4,16], but a major challenge has been to develop a systematic pipeline aimed at discovering growth-relevant TF combinations.

Here we developed a bioinformatics framework to detect cooperative TF promotion of axon growth. The framework centers on the concept of TF co-occupancy, in which functional interactions between TFs are detected by virtue of shared binding to common sets of regulatory DNA. We deployed the pipeline to predict TFs that potentially synergize with Klf6 to drive enhanced axon outgrowth. In vitro phenotypic screening confirmed cooperative TF activity for 12 candidates and 2 independent bioinformatic analyses converged to prioritize interest in a core of 3 factors, nuclear receptor subfamily 5, group A, member 2 (Nr5a2), retinoic acid receptor β (Rarb), and Eomes. Systematic tests of forced TF co-expression in vivo using a pyramidotomy model of axon injury revealed strong and consistent promotion of corticospinal tract (CST) axon growth by combined expression of Klf6 and Nr5a2. Finally, transcriptional profiling of purified CST neurons revealed transcriptional correlates to the evoked growth, notably gene modules related to biosynthesis and DNA repair. Overall, we have identified novel combinations of TFs that drive enhanced axon growth in mature CNS neurons and provide a generalized computational roadmap for the discovery of cooperative activity between TFs with potential application to a wide range of neural activities.

## Results

**In vitro screening identifies TFs that synergize with Klf6 to drive increased neurite outgrowth.** TFs that functionally synergize often do so by binding DNA in close proximity and initiating complementary transcriptional mechanisms[17–20]. Thus, analysis of TF co-occupancy can reveal instances of cooperation between TFs[21,22]. It was shown previously that the TF Klf6 is expressed in CST neurons during developmental periods of axon growth, is downregulated during postnatal maturation, and that forced re-expression in adult neurons modestly enhances axon growth[7]. We therefore reasoned that during development, additional TFs likely co-occupy regulatory elements alongside Klf6 and cooperate to regulate transcription. We further hypothesized that identifying these co-occupiers and supplying them to adult neurons along with Klf6 may lead to further enhancement of axon growth after injury.

To identify potential Klf6 co-occupiers, we first assembled a list of genes that decline in expression as cortical neurons mature, using data from corticospinal motor neurons (CSMNs) in vivo and cortical neurons ages in vitro[23,24] (Fig. 1a). Genes common to both datasets were selected for further analysis. To tighten the focus on genes that contribute to axon growth, we selected those genes associated with Gene Ontology (GO) terms linked to axon growth, including cytoskeleton organization, microtubule-based transport, nervous system development, and regulation of growth (Fig. 1b). Supplementary Data 1 summarizes the final list of 308 developmentally downregulated pro-growth genes and their associated GO terms. For each gene, promoters were assigned as 1500 bp upstream/300 bp downstream of the transcription start site (TSS). To assign enhancers, we took advantage of recent advances in machine learning-based algorithms, deploying the activity-by-contact (ABC) pipeline to reconstruct functional enhancer–gene pairs during active periods of developmental axon growth[25]. Utilizing open-access ENCODE datasets of chromatin accessibility, H3K27Ac enrichment, HiC, and expression datasets from embryonic forebrain, ABC identified a total of 1230 enhancers linked to the selected growth-relevant genes[26] (Fig. 1c). Each gene was associated with those between 1 and 13 enhancers (mean $6 \pm 3$ SEM), 80% of which were located 10–1500 kb from the TSS. The full list of enhancer–gene pairs with genomic coordinates is summarized in Supplementary Data 1. Finally, to predict the binding of Klf6 and potential co-occupying TFs, we examined promoters and enhancers using an anchored TF motif analysis algorithm. This algorithm scans sequences for the canonical Klf6-binding motif and then identifies TF-binding motifs that are over-represented in nearby DNA[27]. The result was 62 candidate TFs that were predicted to frequently co-occur with Klf6 in pro-growth promoters, pro-growth enhancers, or both (Fig. 1d).

We next used high content screening to test the prediction that these candidate TFs functionally synergize with Klf6 to enhance neurite outgrowth (Fig. 2a). Using a well-established screening platform, postnatal cortical neurons received candidate genes by plasmid electroporation and were cultured at low density on laminin substrates, followed 2 days later by automated tracing to quantify neurite outgrowth[7,12,24,28–31]. Nuclear-localized enhanced green fluorescent protein (EGFP) served to mark transfected neurons and βIII tubulin immunohistochemistry labeled neuronal processes for automated tracing (Fig. 2b). All screening plates included wells with three standard treatments as follows: (1) *EBFP* plasmid alone, (2) *Klf6* mixed with mCherry control, and (3) *Klf6* mixed with *VP16-Stat3*, a combination that was shown previously to elevate neurite length above *Klf6* alone[7]. This design accounts for inter-plate variability by normalizing all lengths to *EBFP*, while establishing on each plate both the *Klf6*-only effect size and the sensitivity to potential increase above that level. As expected, across the screen forced expression of *Klf6* alone increased neurite outgrowth by 28% ($\pm$4.37% SEM) compared to *EBFP* control, whereas combination with *VP16-Stat3* further elevated lengths to 61% ($\pm$5.59%) of *EBFP* ($p$-value < 0.01, analysis of variance (ANOVA) with post hoc Fisher's least significant difference (LSD)). The remainder of each plate was devoted to *Klf6* mixed with candidate TFs, which were delivered

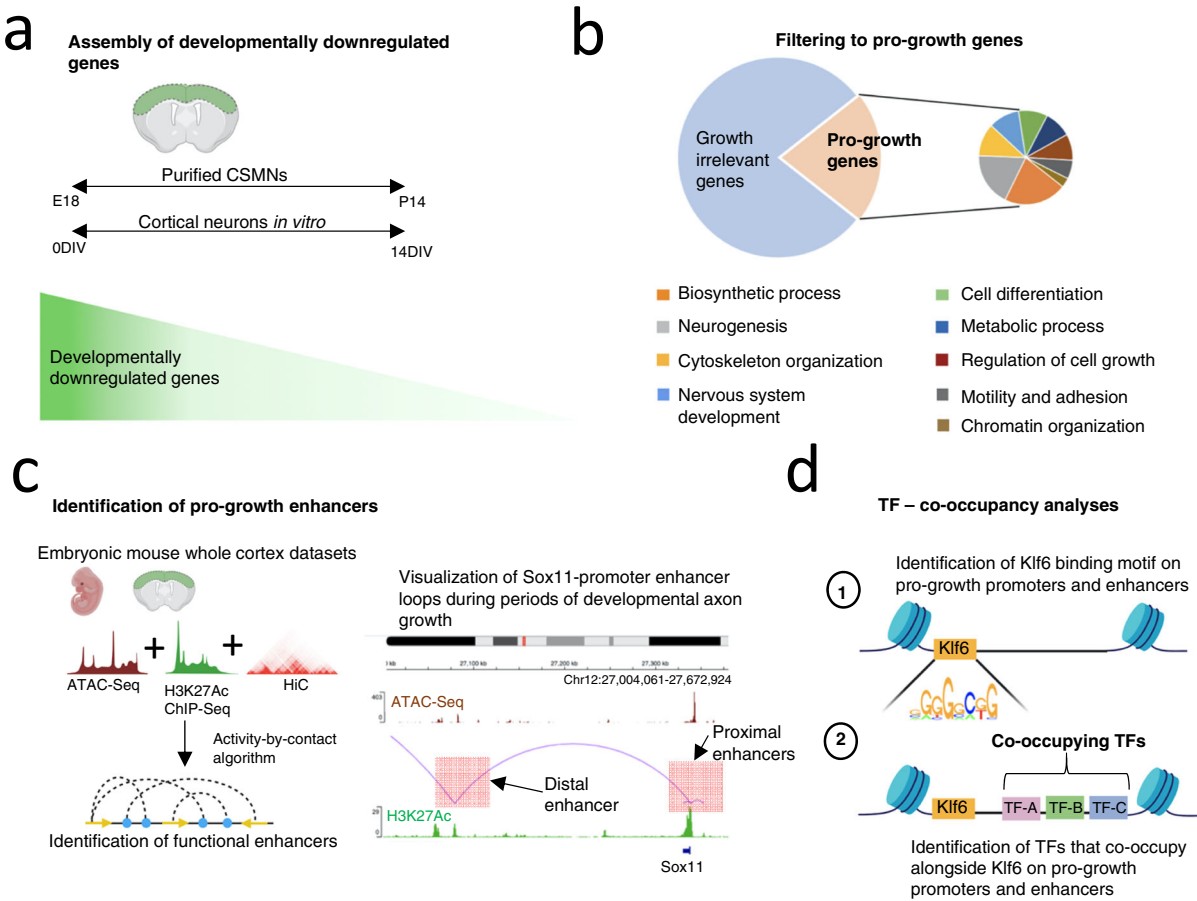

**Fig. 1 Integrated bioinformatics pipeline identifies KLF6 partner TFs likely involved in the regulation of axon growth. a** Time-course gene expression datasets from mouse cortical neurons were integrated to isolate genes that are downregulated across cortical maturation in vivo and in vitro (CSMNs, corticospinal motor neurons). **b** Gene Ontology analyses delineated developmentally downregulated genes that are likely growth relevant, based on functional enrichment in GO categories pertinent to axon growth. **c** Activity-by-contact (ABC), an enhancer–target gene pairing algorithm, defined specific enhancer regions genomewide, which modulate the expression of growth-relevant transcripts. Genome tracks are loaded to visualize the proximal and distal enhancer–promoter loops involved in the regulation of the pro-growth gene *Sox11* during periods of developmental axon growth. ATAC-Seq profiles are shown in magenta and H3K27Ac profiles are shown in green. **d** TF co-occupancy analyses predicted TFs that co-occupy regulatory DNA alongside Klf6 to regulate the expression of growth-relevant transcripts.

in their native form, not modified with VP16, to probe endogenous activity. Each combination was tested in three separate experiments and a minimum of 150 individual cells quantified per experiment. Notably, of the 62 candidate TFs, 12 significantly elevated neurite length above the level of Klf6 alone ("hard hits") ($p$-value < 0.05, ANOVA with post hoc Fisher's LSD) and an additional five missed statistical significance but improved growth in two of three replicate experiments ("soft hits") (Fig. 2a). Detailed screening data are available in Supplementary Data 2 and Supplementary Fig. 1 provides data for the effects of TFs on additional morphological parameters. Interestingly, all 12 TFs that elevated neurite length came from the one-third of candidates that were predicted to co-occupy with Klf6 in both promoters and enhancers, and no hits came from TFs predicted to co-occupy solely in promoters or in enhancers (Fig. 2c). These data support the ability of co-occupancy analysis to predict functional interactions between TFs and emphasize the importance of scanning both the promoter and enhancer elements.

We then used network analyses to prioritize interest in the 12 "hard hit" TFs. First, using open-access TF-binding data, top transcriptional targets of the 12 hit TFs were used to generate individual TF target gene networks. These networks were then merged, expanded to bring in immediately connected genes based on known interactions, and restructured to move genes with maximal connections to the center and minimal connections to the periphery. This resulted in a unified network with a core of TFs predicted to exert maximal influence, surrounded by shells of TFs with progressively lower connectivity (Fig. 2c). Notably, during the expansion step, Klf6 itself was among the genes that were spontaneously added to the network. Klf6 was the only added TF to be placed in the core, even as additional pro-growth TFs including Sox11, Stat3, Myc, and Jun appeared in inner shells. The appearance of these TFs and their central positions in the network substantiates the premise that the constructed network is Klf6-centered and relevant to axon growth[2,7–11,32–34]. Moreover, all five of the "soft hit" TFs also appeared but were placed in outer shells of the network, hinting at interactions near the threshold of screening sensitivity. Most importantly, three hit TFs (Nr5a2, Eomes, and Rarb) occupied the core with Klf6, prioritizing subsequent interest. Thus, an integrated pipeline of co-occupancy analyses, high content screening, and network analyses point toward three TFs as potentially cooperative with Klf6.

For independent validation, we performed a second analysis, this time starting from a previously published set of genes that are upregulated in cortical neurons in vitro upon forced expression

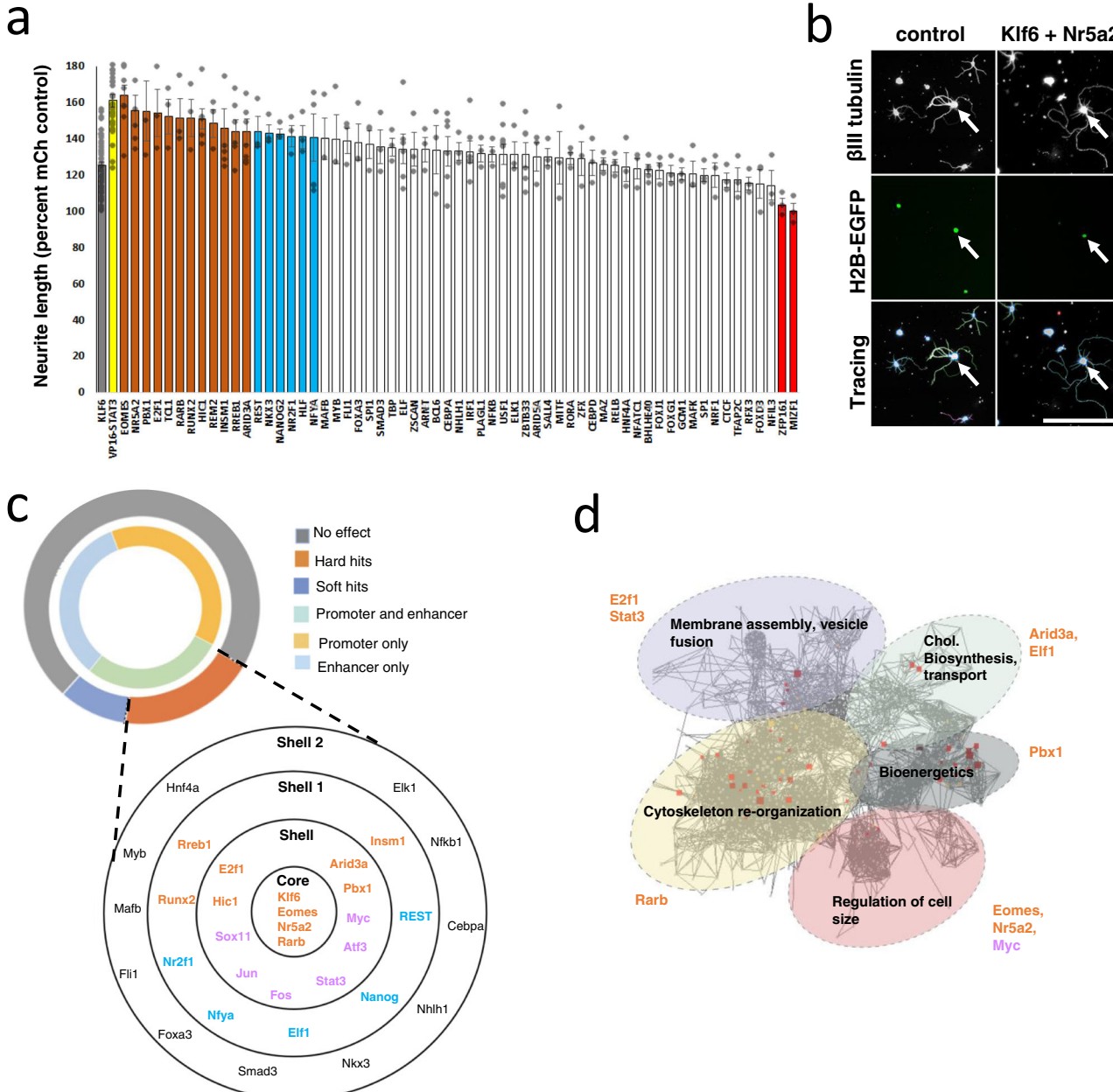

**Fig. 2 In vitro screening and network analyses identify a novel pro-growth transcription factor network. a** Postnatal cortical neurons were transfected with control plasmid, Klf6 with control, Klf6 with VP16-Stat3, or Klf6 combined with 62 candidate TFs. After 2 days in culture, automated image analysis (Cellomics) quantified neurite lengths normalized to the average of on-plate controls. The solid black line indicates the average control length. Combined expression of Klf6 and Vp16-Stat3 showed expected increase in neurite length above Klf6 alone, confirming assay sensitivity (yellow bar, *p*-value < 0.0001, ANOVA with post hoc Fisher's LSD). Twelve candidate TFs significantly enhanced neurite outgrowth when combined with Klf6 (orange bars, *p*-value < 0.05, ANOVA with post hoc Fisher's LSD) and five more enhanced growth in two of three replicates but missed the cutoff for statistical significance (blue bars). *n* > 150 cells in each of three replicate experiments with similar results; exact cell numbers and *p*-values are available in Supplementary Data 2. **b** Representative images showing transfected neurons (white arrows, H2B-EGFP) and neurite tracing. **c** Network analysis was performed on transcripts upregulated by Klf6 overexpression in culture neurons, followed by motif analysis of gene promoters within each module. Of the ten TF motifs with the highest enrichment, seven were hit TFs from the screening experiment (orange), including Rarb, Eomes, and Nr5a2. **d** Hit TFs were used to build networks. All hit TFs derived from candidates with predicted Klf6 co-occupancy in both promoters and enhancers. Rarb, Eomes, and Nr5a2 were located in the network core, where Klf6 itself also appeared during network construction. Several soft hit TFs (blue) and established pro-growth TFs (pink) spontaneously populated outer shells. Scale bar is 100 μm, error bars indicate SEM. Source data are provided as a Source Data file.

of Klf6[7]. We again performed a network analysis of Klf6-upregulated genes, constructing functionally related subnetworks, and then examined gene promoters within each subnetwork for the enrichment of TF motifs. This independent analysis detected enrichment for the motifs of seven factors, which remarkably

included the three "core" TFs identified above (Fig. 2d). Thus, genes that respond to Klf6 overexpression are enriched for the recognition motifs of Eomes, Nr5a2, and Rarb, supporting the hypothesis that they cooperate with Klf6 to activate pro-growth gene networks.

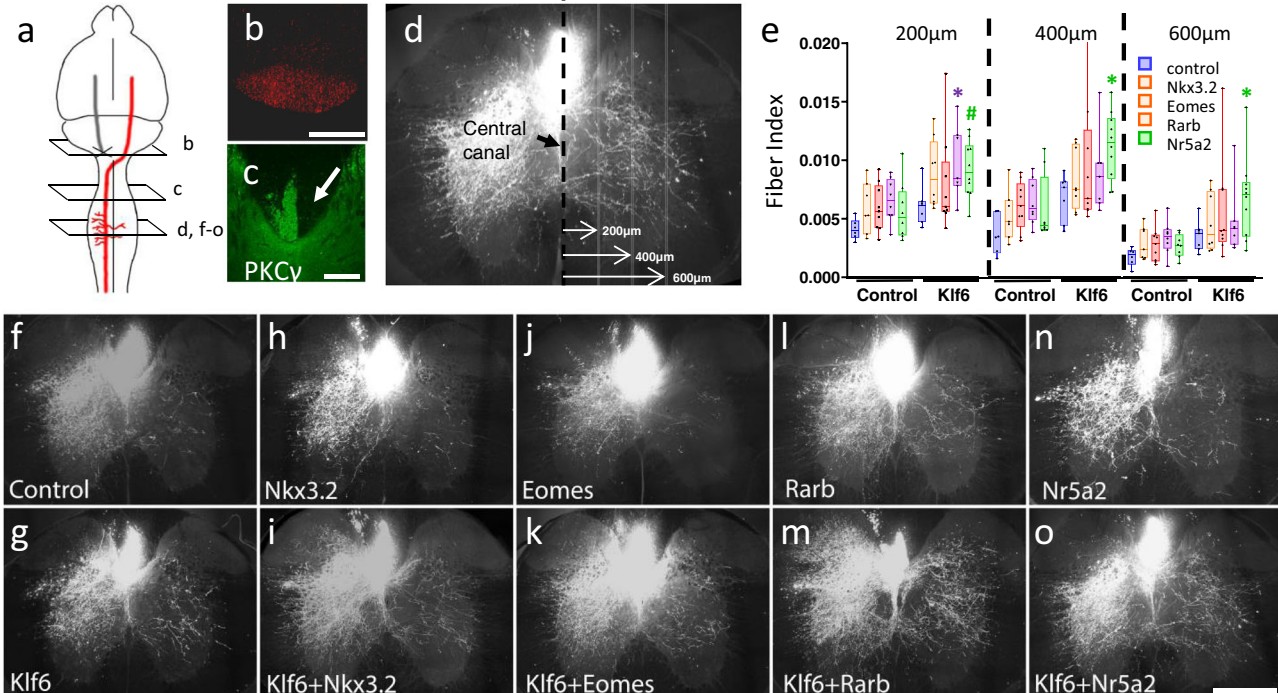

**Fig. 3 Nr5a2 synergizes with Klf6 to promote CST growth. a** Adult mice received cortical AAV delivery of single or combinatorial TFs and contralateral pyramidotomy. **b** Co-injected tdTomato-labeled CST axons, visible in cross-section of the medullary pyramids. **c** PKCγ immunohistochemistry confirmed unilateral ablation of the CST (arrow). **d, e** Cross-midline growth of CST axons was quantified in transverse sections of cervical spinal cord by counting intersections between labeled axons and virtual lines at 200, 400, and 600 μm from the midline, normalized to total labeled axons counted in the medulla (Fiber Index). Candidate TFs administered singly did not significantly elevate axon growth, whereas Nr5a2 and Rarb elevated growth above the level of Klf6 alone. [#$p = 0.0502$, *$p < 0.05$ (K + R 200 μm–0.0284, K + N 400 μm–0.0129, K + N 600 μm–0.0367) two-way ANOVA, post hoc Dunnett's]. **f–o** Representative images of cervical spinal cord showing elevated cross-midline growth in animals treated with Klf6 and Nr5a2. Scale bar is 500 μm. *n* is between six and ten animals for all groups; PKCγ, medulla, and spinal images from all animals are in Supplementary Figs. 2–5. Box plots represent median and IQR (25th to 75th percentile), and whiskers extends to the maximum and minimum values. Dots in **e** represent values of single animals. Source data are provided as a Source data file. Scale bars are 200 μm (**b, c**) and 500 μm (**f–o**).

**Nr5a2 synergizes with Klf6 and Rarb to drive enhanced CST sprouting following pyramidotomy injury.** We next asked whether forced expression of the three hit TFs, singly or in conjunction with Klf6, can promote the growth of CST axons in vivo. Based on serotype availability, we first verified the ability of AAV2-Retro vectors, previously shown to effectively transduce CST neurons in a retrograde manner[35], to also transduce neurons when applied directly to cell bodies. We delivered a titer-matched mixture of AAV-H2B-mEGFP and AAV-H2B-mScarlet to the cortex of adult mice, followed by retrograde labeling of CST neurons by cervical injection of CTB-647. Two weeks later, 3.6% (±1.5%) of transfected CST neurons expressed only EGFP, 1.0% (±0.3%) expressed only mScarlet, and 95.4% (±1.5%) expressed both fluorophores (Supplementary Fig. 2). Accordingly, TFs were delivered to adult mice by cortical injection of Retro-AAV (hereafter shortened to AAV), with AAV-Cre acting as control and also included in single-gene treatments to equalize the viral load in all animals (see Supplementary Data 3). tdTomato tracer was co-injected in serotype AAV9, avoiding potential complications of retrograde spread of the tracer itself. Mice received unilateral pyramidotomy, followed 8 weeks later by quantification of cross-midline sprouting of corticospinal (CST) axons in the cervical spinal cord, measured at 200, 400, and 600 μm from the midline (Fig. 3a, d). Axon counts were normalized to tdTomato+ axons in the medullary pyramids (Fig. 3b) and injuries were confirmed by unilateral ablation of PKCγ in spinal sections (Fig. 3c). Raw images of all animals are provided in Supplementary Figs. 3–5. In addition to Cre, we also included Nkx3.2, a

non-hit/non-core TF, as an additional negative control; RNAscope or immunohistochemistry confirmed overexpression of all TFs (Supplementary Fig. 6a–e). As expected, Klf6 expression significantly increased normalized axon counts at all distances across the midline ($p < 0.01$, two-way ANOVA) (Fig. 3e, g). When expressed singly, none of the candidate TFs significantly elevated axon growth above the level of control (Fig. 3e, h, j, l, n). When combined with Klf6, however, significant effects emerged (Fig. 3e–o). Most notably, co-expression of Nr5a2 with Klf6 consistently elevated axon counts above that of Klf6 alone and above the maximum growth in Cre controls in all animals. Co-expression of Rarb with Klf6 drove growth that exceeded Klf6 alone at 200 μm from the midline (Fig. 3e). In three of nine animals, combined Eomes/Klf6 expression yielded high axon counts, but this effect was inconsistent and did not reach statistical significance (Fig. 3e). Finally, as expected, Nkx3.2 produced no significant elevation in axon growth above the level of Klf6 alone (Fig. 3e, i). In summary, these data identify robust synergy between Klf6 and candidate TFs, most notably Nr5a2.

We then performed a second in vivo pyramidotomy experiment to confirm Nr5a2's synergy with Klf6 and to determine whether Nr5a2 might similarly potentiate the growth of other pro-growth TFs. AAV delivery, injury, and axon quantification were identical to the first experiment and detailed histology is available in Supplementary Figs. 3–5. In this experiment, Nr5a2 acted as the base gene, to which control, Klf6, or Rarb were added. Consistent with the prior experiment, combined Nr5a2/Klf6 expression elevated CST growth above the level of either

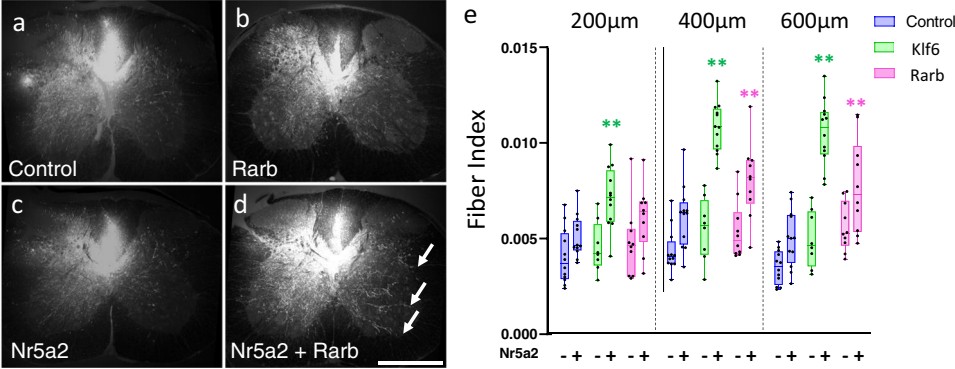

**Fig. 4 Nr5a2 synergizes with Rarb to promote CST growth. a–d** Adult mice received cortical AAV delivery of single or combinatorial TFs and contralateral pyramidotomy. Co-injected AAV-tdTomato-labeled CST axons. Elevated cross-midline sprouting of CST fiber is evident in animals treated with combined Nr5a2 and Rarb (arrows, **d**). **e** Cross-midline growth of CST axons was quantified in transverse spinal sections. Nr5a2 administered alone had no significant effect on axon growth (blue bars), but increased axon growth when combined with either Klf6 or Rarb, significantly above the level of either alone (green and pink bars, respectively). [**$p < 0.01$ (K + N 200 μm = 0.0012, K + N 400 μm = 0.0001, K + N 600 μm = 0.0001, K + R 400 μm = 0.0007, K + R 600 μm = 0.0136, two-way ANOVA, post hoc Dunnett's]. Scale bar is 500 μm. $n$ is between 10 and 14 animals per group. PKCγ, medulla, and spinal images from all animals are in Supplementary Figs. 2–5. Box plots represent median and IQR (25th to 75th percentile), and whiskers extends to the maximum and minimum values. Dots in **e** represent values of single animals. Source data are provided as a Source data file.

alone (Fig. 4e). Moreover, axon growth in the lowest Klf6/Nr5a2 animal still exceeded that in the maximum control animal, illustrating the strength and consistency of the effect. Interestingly, combined Nr5a2 and Rarb also elevated CST growth above the level of either alone (Fig. 4a–e). This effect did not appear as striking as that of Nr5a2/Klf6, but nevertheless substantiates the ability of network analysis to predict functional TF cooperation. Thus, combinations of TFs, rationally selected based on the basis of predicted co-occupancy of pro-growth regulatory DNA, can enhance CST axon growth in vivo.

Next, focusing on the Klf6 and Nr5a2 combination, we performed additional experiments to substantiate and clarify the phenotypic effects in vivo. First, to further confirm successful co-expression, we performed additional cortical injections, using the same viral loads as the pyramidotomy experiments, followed by RNAscope-based co-detection of Klf6 and Nr5a2. In animals that received AAV-tdTomato with AAV-Klf6 and AAV-Nr5a2, >90% of tdTomato-positive cells showed a strong label with both transgenes (Supplementary Fig. 6). Thus, tdTomato-labeled axons in the spinal cord in Nr5a2/Klf6 animals largely arose from CST neurons expressing both TFs. We next compared inflammation, gliosis, and cell death in the cortex of animals that received control vs. Klf6/Nr5a2. Three days after injection, CD11b, Glial Acidic Fibrillary Protein (GFAP), and Terminal Deoxynucleotidyl Transferase dUTP Nick End Labeling (TUNEL) reactivity near the needle tracks were similar across treatments (Supplementary Fig. 7a–d, i, j). We also examined cortices in which CST neurons were identified by retrograde labeling and again detected similarly low levels of GFAP, CD11B, and TUNEL reactivity (Supplementary Fig. 7e–h, i, j). These data support a model in which Klf6/Nr5a2 act directly in CST neurons to influence axon growth, as opposed to acting indirectly by influencing gliosis, inflammation, or cell death.

We next re-examined the morphology of Klf6/Nr5a2-evoked growth by quantifying the frequency of branch points in CST axons. Starting with tissue sections from the second pyramidotomy experiment, axons that sprouted across the cervical midline were visualized at high magnification by confocal microscopy (Fig. 5a–d). Segments of axons that remained continuously within the plane of visualization for a minimum of 100 μm were identified and traced, and the number of definitive branches along the segment was normalized to the traced length. Interestingly,

CST axons in Klf6/Nr5a2-treated animals displayed significant elevation of the frequency of branch formation (Fig. 5e–g control: 2.5 ± 0.4 SEM branches/mm, Klf6/Nr5a2: 9.6 ± 0.5 SEM branches/mm). This finding is consistent with the in vitro screening data, which showed that Klf6/Nr5a2 expression increased the length of the longest neurite when branches were included in the tracing, but not when only the longest unbranched path was analyzed (Supplementary Fig. 1d, e). Combined, these data indicate that combined Klf6/Nr5a2 expression in cortical neurons can act to increase branch formation and growth.

Finally, we tested the effects of forced expression of Klf6/Nr5a2 in a model of direct axon injury. Adult mice received cortical injections of AAV-EGFP tracer and either AAV-Cre control or combined AAV-Klf6 and AAV-Nr5a2. Animals received a severe crush injury to the thoracic spinal cord, followed 4 weeks later by perfusion and visualization of axons in horizontal spinal sections (Fig. 6a–d). GFAP reactivity defined the injury site, which, as expected, spanned the entirety of the cord in all animals. In control and Klf6/Nr5a2-treated animals alike, CST axons were completely interrupted by the injury and in no animals did we observe an extension of axons from the severed ends into or beyond the injury site. The distance between the lesion edge and CST axons was similar between groups, suggesting similar retraction behavior. Notably, however, axons treated with Klf6/ Nr5a2 displayed a strong sprouting response into spinal tissue rostral to the injury, including robust extension across the spinal midline. Individual axons were clearly visualized crossing the midline and Kfl6/Nr5a2 animals showed a significant elevation of axon density in contralateral spinal cord, normalized to axon counts in the medulla (Fiber Index, Fig. 6e). Thus, although Klf6/ Nr5a2 does not confer to CST axons an ability to traverse the strongly inhibitory environment of a spinal lesion, it does act in injured axons to increase the propensity for growth by axon collaterals.

**Combined Klf6-Nr5a2 treatment leads to the upregulation of gene modules related to macromolecule biosynthesis and DNA repair.** To probe the underlying mechanisms of growth promotion, we profiled the transcriptional consequences of single and combined expression of Klf6 and Nr5a2 in CST neurons. To

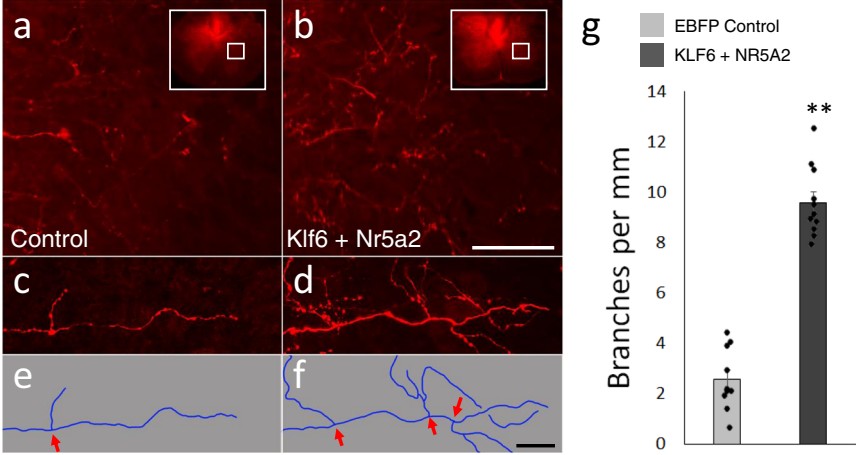

**Fig. 5 Combined Klf6/Nr5a2 treatment increases the frequency of CST axon branching. a, b** CST axons transduced with control (**a**) or Klf6 and Nr5a2 (**b**), which have extended across the midline into contralateral cervical spinal cord. **c–f** At higher magnification, instances of branching can be identified (arrows). **g** Animals that received combined Klf6 and Nr5a2 showed a significant increase in the frequency of branch formation above control animals ($p = 1.89^{-10}$, two-tailed $t$-test). A minimum of 5 mm of growth, from three separate sections, was quantified for each of ten animals in both groups. Scale bars are 100 μm (**a**, **b**) and 20 μm (**c–f**). Dots in **g** represent values in single animals. Source data are provided as a Source data file.

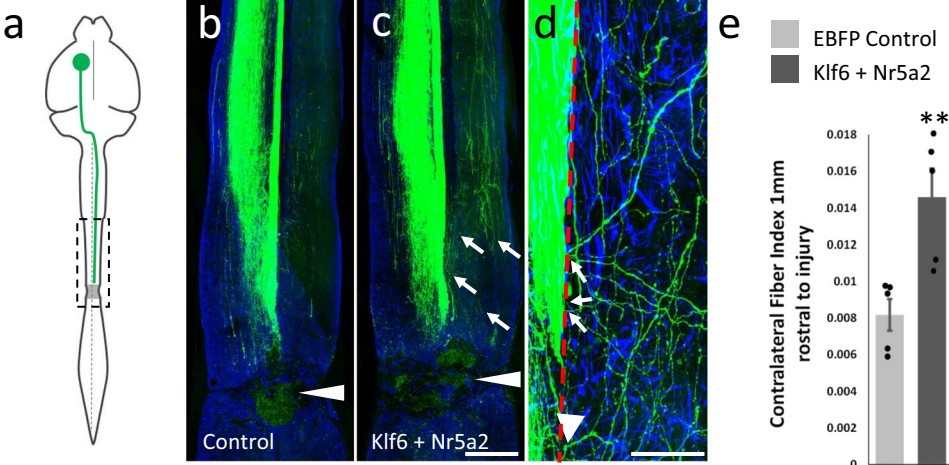

**Fig. 6 Combined expression of Klf6 and Nr5a2 in corticospinal neurons increases collateral sprouting but not extension through sites of spinal lesion. a** Adult mice received cortical injection of AAV-EGFP tracer with AAV-Cre control or combined AAV-Klf6 and AAV-Nr5a2, and T10/11 spinal crush injuries. **b, c** Horizontal spinal sections with examples of CST axons (green), spinal injuries (arrowheads), and reactive astrocytes (GFAP, blue). Neither EBFP control nor KLF6/Nr5a2-treated axons traverse the injury, but Klf6/Nr5a2-treated axons display cross-midline sprouting rostral to the injury (arrows). **d** CST collaterals crossing the midline (dotted line) and extending into contralateral cord. **e** Quantification of CST axons that extend 400 μm from the midline, normalized to total CST axons counted in the medullary pyramids, shows a significant elevation in cross-midline growth (**$p = 0.0069$, two-tailed $t$-test, $n = 5$ control, 5 Klf6/Nr5a2). Scale bars are 1 mm (**b, c**) and 100 μm (**d**). Dots in **e** represent values of single animals. Source data are provided as a Source data file.

isolate CST neurons, nuclear-localized fluorophores and TFs were expressed by cervical injection of Retro-AAV2, which results in highly efficient and selective transgene expression by retrograde transduction[35] (Fig. 7a, b). Animals received pyramidotomy injury, followed 1 week later by purification of labeled nuclei of CST neurons from the spared hemisphere by flow cytometry, isolation of messenger RNA, and construction and sequencing of RNA libraries. Differential gene expression was determined using EdgeR software[36]. Compared to tdTomato control, single expression of Klf6 resulted in the upregulation of 208 transcripts (transcripts with fragments per kilobase of transcript per million mapped reads (FPKM) > 5 and log2fold change>1; $p$-value < 0.05, false discovery rate (FDR) < 0.05) (Fig. 7c). Reminiscent of prior findings in cultured neurons[7], network analysis of Klf6-responsive genes revealed subnetworks with functions highly relevant to axon growth, including CNS development, neuron projection development, migration, adhesion, and cytoskeleton organization (Fig. 7d–f). Indeed, neuron projection and cytoskeleton networks comprised >60% of Klf6-responsive genes, consistent with the effects of Klf6 overexpression on axon growth (Fig. 7d–f). Single overexpression of Nr5a2 had little unique effects on gene expression, with only 12 transcripts identified as upregulated only in the Nr5a2 group and 130 transcripts shared with the Klf6 treatment group (Fig. 7c). Dual Klf6/Nr5a2 expression, however, drove upregulation of 192 transcripts that were not significantly increased by either alone (Supplementary Data 4 and Fig. 7c). Interestingly, network analysis of these transcripts revealed modules involved in macromolecule biosynthesis, DNA repair, and development, all of which showed substantial increases in the net expression above control levels (Fig. 7g–i).

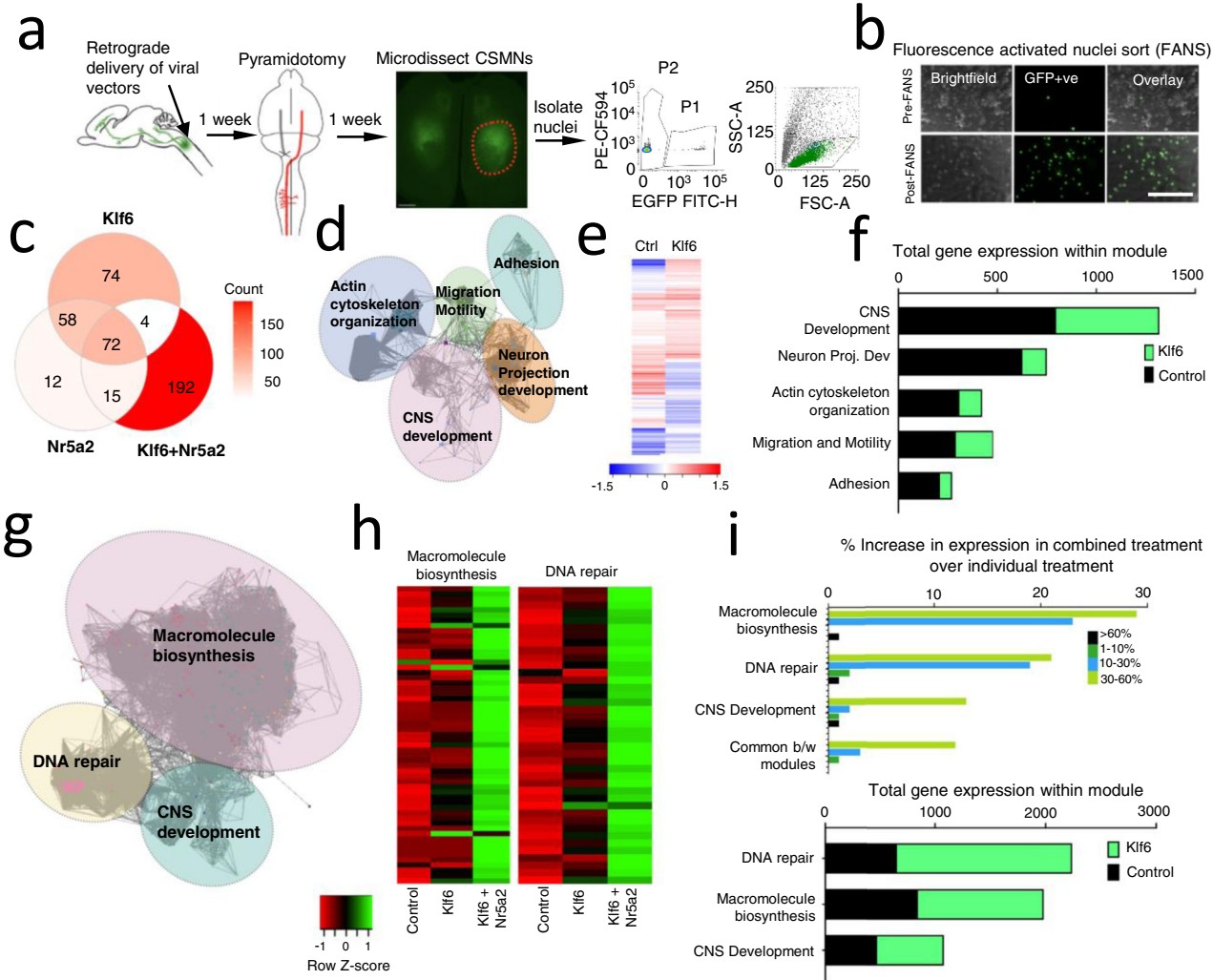

**Fig. 7 Combined Klf6, Nr5A2 treatment induces expression of gene modules involved in macromolecule biosynthesis and DNA repair. a** Overview of sample collection for RNA-Seq analysis. (CSMNs, corticospinal motor neurons). **b** Nuclei before and after FANS, confirming successful purification of mNeongreen+ nuclei. **c** Venn diagram showing transcript comparison across the three gene treatments. **d** Regulatory network analysis of genes upregulated after Klf6 overexpression revealed subnetworks enriched for distinct functional categories highly relevant to axon growth. Nodes correspond to target genes and edges to multiple modes of interaction (physical, shared upstream regulators, shared signaling pathways, and inter-regulation). Only significantly enriched GO categories were included in the network analysis ($p < 0.05$, right-sided hypergeometric test with Bonferroni correction, Cytoscape). **e** Heatmap showing selection of differentially expressed genes between control and Klf6 treatment. **f** The number of genes within each functional subnetwork (top) and the total FPKM values in each in control and Klf6 overexpression groups. **g** Regulatory network analysis of genes uniquely upregulated after combined Klf6 and Nr5a2 overexpression (not upregulated in single treatments) revealed subnetworks enriched for functions in macromolecule biosynthesis and DNA repair ($p < 0.05$, right-sided hypergeometric test with Bonferroni correction, Cytoscape). **h** Heatmap showing selection of differentially expressed genes in DNA repair and macromolecule biosynthesis modules. Genes that are upregulated (green) and downregulated (red) relative to control are shown. Color values indicate average FPKM expression values as shown in the legend. Columns correspond to treatment groups and rows to genes. **i** The percent change in FPKM values within each module, revealing that combined Klf6 and Nr5a2 treatment most affect genes in DNA repair and macromolecule biosynthesis modules (top). Graph shows the number of genes within functional subnetworks (bottom). $n = 3$ animals pooled per replicate, 2 replicates/gene treatment with similar results (Supplementary Data 4). Scale bar is 100 μm. Source data are provided as a Source Data file.

To further validate the findings of this bulk RNA sequencing (RNA-Seq) approach, we performed a replicate study, this time employing a single-nuclei RNA-Seq approach. Nuclear-localized fluorophores and TFs (Control or Klf6 + Nr5a2) were expressed by cervical injection of Retro-AAV2 as described above (Supplementary Fig. 8a) Animals received pyramidotomy injury, followed 1 week later by purification of labeled nuclei of CST neurons by flow cytometry, construction, and sequencing of single-nuclei libraries. Using this approach, we sequenced and analyzed ~3000 control nuclei and Klf6/Nr5a2-treated nuclei (Supplementary Fig. 8b). The majority of nuclei partioned into

clusters highly enriched for established markers of CST neurons, including Bcl11b, Cdh13, Tmem163, Crim1, Cntn6, and Slco2a1, indicating successful purification of CST neurons[23,37,38] (Supplementary Fig. 8c). We integrated control and Kfl6/Nr5a2 single-nuclei datasets using SEURAT[39], and carried out differential gene expression analyses to identify Klf6/Nr5a2-responsive target genes, specifically in the high-confidence CST cluster (Supplementary Data 4). As expected, Klf6 and Nr5a2 themselves were detected as strongly upregulated. In addition, when comparing the output of the initial bulk RNA-Seq and single-nuclei RNA-Seq, 144 genes were commonly called as significantly regulated by

Klf6/Nr5a2 in both the datasets, 101 of which were shared with the 192 transcripts called as Klf6/Nr5a2-specific in the bulk RNA-Seq experiment (>50% overlap) (Supplementary Fig. 8d). Importantly, all shared target genes were located in clusters functionally linked to roles in Macromolecule Biosynthesis, DNA repair, and CNS development, providing independent support for the regulation of these processes by combined experssion of Kfl6 and Nr5a2 (Supplementary Fig. 8e–g). Intriguingly, recent analyses of regeneration-competent peripheral neurons also showed that genes involved in macromolecule biosynthesis and DNA repair are upregulated in the course of successful regeneration[14], supporting their potential involvement in axon growth. In aggregate, these data substantiate the ability of forced Klf6 expression to activate gene networks relevant to axon growth and point toward the ability of dual Klf6/Nr5a2 expression to activate gene networks involved in macromolecule synthesis and DNA repair as a likely explanation for their combined enhancement of axon growth.

## Discussion

Using interlocking bioinformatics approaches, high content screening, and in vivo testing, we have identified TF combinations that synergize to enhance CST axon growth. In addition, transcriptional profiling and network analyses provide molecular correlates to the evoked responses. These findings clarify cellular functions that drive axon extension and serve as a roadmap for ongoing efforts to discover combinatorial gene treatments to promote regenerative axon growth.

Klf6 belongs to the KLF family of TFs, which have well-studied roles in regulating axon growth[7,12,28,40]. Both Rarb and Nr5a2/LRH-1 belong to the nuclear receptor family of TFs, with diverse functions in development and cellular metabolism[41]. Rarb has previously been linked to axon growth and a Rarb agonist is under evaluation for clinical efficacy in a brachial plexus avulsion model of injury[42]. In contrast, Nr5a2 is unstudied in the context of axon extension. It has, however, been linked to the differentiation of neural stem cells and to cellular reprogramming, and can replace Oct4 in the reprogramming of murine somatic cells to pluripotent cells[43,44]. Our present work adds to the growing body of evidence that TFs involved in neuronal differentiation may positively influence axon growth[9,13].

An important finding is that although Nr5a2 does not improve axon growth when expressed individually, it strongly potentiates the effect of overexpressed Klf6. The importance of combinatorial gene activity for axon growth, as opposed to single-factor effects, is increasingly recognized[2,4,45]. Regeneration-competent cell types such as zebrafish retinal ganglion cells and peripherally injured sensory neurons respond to axotomy by upregulating groups of TFs that likely synergize to drive regenerative axon growth[14,15,46]. An unmet challenge, however, has been to turn the conceptual appreciation for the role of TF synergy into an operational workflow to discover effective gene combinations for axon extension in the injured nervous system. Whereas prior efforts have focused on direct protein–protein interactions to predict TF synergy[46], here we exploited the concept of TF co-occupancy. A search for factors with recognition motifs in proximity to those of Klf6 yielded a set of 62 candidate TFs, which was further narrowed by phenotypic screening. A key observation from the cell culture experiments is that none of the hit TFs were predicted to co-occupy solely in promoter regions; rather, all hit TFs were characterized by predicted co-occupancy in both promoters and enhancers. This suggests a key role for enhancer elements in regulating pro-growth gene transcription, in line with the recent finding in Dorsal Root Ganglion (DRG) neurons that modulation of enhancers may be critical for transcriptional activation of growth genes[13,47].

Finally, guided by network analysis of the screening results, a systematic campaign of combinatorial testing in vivo identified highly consistent enhancement of CST axon growth by combined expression of Klf6 and Nr5a2. Importantly, in a model of pyramidotomy injury, delivery of Kfl6 and Nr5a2 did not affect cell survival but did affect both the overall density and frequency of branching of collateral axons in contralateral spinal cord. Moreover, in axotomized CST axons, forced expression also increased the growth of collateral branches, although not regenerative growth across a spinal lesion. Importantly, in contrast to prior reports of CST regeneration through crush sites, injuries here were performed using double crushes with wider forceps, thus producing complete lesions without astrocytic bridges[48,49]. The absence of regenerative advance in this model indicates that treated axons remained sensitive to environmental inhibition at the lesion, but do not rule out the possibility that Klf6 and Nr5a2 treatment could produce regenerative growth into more favorable environments, e.g., progenitor-derived tissue grafts[50]. In addition, the strong effects of combined Klf6 and Nr5a2 on collateral growth in both spared and injured CST axons suggests that this treatment could be highly beneficial in situations of partial injury by enhancing increased branch elaboration by spared axons and potentially fostering relay circuity by injured axons[51,52]. A critical question for future research is whether the stimulated branches succeed in forming functional synapses on target cells in the spinal cord. We believe this evidence of TF synergy in driving growth of axons following CNS injury in vivo is a critical advance for the field. Although these experiments were anchored on Klf6, these data show a pipeline that can be re-deployed to identify additional TF combinations that synergize to drive growth, or potentially other cellular functions.

A key insight from the current work is to clarify in vivo, in purified CST neurons, the transcriptional events that are triggered by Klf6, Nr5a2, and the two together. Forced expression of Klf6 activates highly interconnected groups of genes with functions in various aspects of axon growth spanning terms such as Neuron Projection Development, Cytoskeleton organization, Motility, and Adhesion. We have previously shown that gene modules involved in cytoskeleton remodeling and motility are evoked in response to Klf6 overexpression in cortical neurons in vitro[7], consistent with our in vivo findings here. This overlap underscores the utility of in vitro screening approaches in delineating transcriptional networks relevant to axon growth. Importantly, this finding illustrates the potential of TF-based interventions to trigger broad changes in gene expression; it is most likely that improvements in axon growth reflect the net change in a wide set of transcripts, as opposed to acting through any single target. Combined Klf6/Nr5a2 expression, which led to large and consistent increases in axon growth above those triggered by Klf6 alone, also caused the upregulation of unique modules of genes with functions in macromolecule biosynthesis and DNA repair. These modules were activated by neither Klf6 nor Nr5a2 alone, suggesting that enhanced biosynthesis and/or DNA repair contribute to the enhanced growth in Klf6/Nr5a2-treated animals. A role for the biosynthesis of macromolecules is highly plausible, as active growth depends on re-synthesis of cellular material. Indeed, regeneration-competent zebrafish retinal ganglion cells and mammalian DRGs also respond to injury by upregulating genes involved in macromolecule biosynthesis[13–15]. Moreover, a very recent study in DRG neurons found that peripheral injury, which triggers axon growth, resulted in a sustained upregulation of genes involved in biosynthesis. In contrast, a central injury that does not trigger regeneration led to eventual downregulation, suggesting that a reduction in biosynthesis pathways may restrict axon growth after spinal injury[53]. Our findings regarding the effects of Klf6/Nr5a2 add to multiple lines of evidence that the

upregulation of genes involved in biosynthesis may be an evolutionarily conserved molecular signature of neurons mounting a successful growth response.

Regarding DNA repair pathways, it is interesting to note that increased cellular metabolism and transcription frequently lead to DNA damage[54,55]. Moreover, disruption of DNA repair machinery in mouse retinal progenitors or cortical progenitors leads to reduced, disturbed trajectories of axon growth and guidance[56,57]. Thus, during development, proper axon growth may require efficient DNA repair machinery. Similarly, peripherally injured DRG neurons upregulate several DNA damage response marker genes and blocking this response leads to reduced neurite outgrowth in vitro and impaired regeneration in vivo[58]. The finding that combined Klf6/Nr5a2 expression activates genes involved in DNA repair suggests the intriguing hypothesis that this repair machinery may aid successful axon regrowth in CNS neurons; future work can address this notion.

In summary, a bioinformatic and screening platform, centered on the concept of TF co-occupancy, has revealed synergy between TFs Klf6 and Nr5a2, which leads to improved axon growth after CNS injury.

## Methods

**Mice and husbandry conditions**. All animal testing and research was carried out in compliance with ethical regulations laid out by the National Institutes of Health (NIH) guide for the care and use of animals, and all experimental protocols involving animals were approved by the Institutional Animal Care and Use Safety committee at Marquette University (protocol number AR-309, AR-314). Mice were bred and raised under a 24 h light–dark cycle with 12 h of light and 12 h of darkness. Ambient temperature was maintained at 22 °C ± 2 °C and humidity between 40 and 60%.

**Identification of developmentally downregulated, growth-relevant transcripts**. All expression datasets used to identify developmentally downregulated genes are deposited at NCBI GEO (In vivo CSMNs data by ref. [23] - GSE2039; In vitro primary neurons aged in culture by ref. [24], SRP151916). In vitro RNA-Seq datasets were processed using the updated Tuxedo suite of tools[24]. Trimmed reads were mapped to the rat reference genome [UCSC, Rat genome assembly: Rnor_6.0] using HISAT2 aligner software (unspliced mode along with–qc filter argument to remove low-quality reads prior to transcript assembly)[59]. Transcript assembly was performed using Stringtie[60] and assessed through visualization on UCSC Genome Browser. Differential gene expression analysis was performed using Ballgown software (Default parameters)[60]. Transcripts were considered significant if they met the statistical requirement of having a corrected $p$-value of <0.05, FDR < 0.05.

For the in vivo CSMN microarray datasets, weighted correlation network analysis was performed[61]. First, relevant microarray datasets (GSE2039) were loaded into R and basic pre-processing of data was performed to remove outlier data and handle probes with missed data. Outlier data were identified by performing hierarchical clustering to detect array outliers. Numbers of missing samples in each probe profile were counted and probes with extensive numbers of missing samples were removed. Next, a weighted correlation network was constructed by creating a pairwise Pearson correlation matrix, which was transformed into an adjacency matrix using a power of 10. Then, topological overlap was calculated to measure network interconnectedness and average linkage hierarchical clustering was done to group genes on the basis of the topological overlap dissimilarity measure. Finally, using a dynamic tree-cutting algorithm, we identified 20 gene modules. Gene modules with developmental downregulation in expression of twofold and above were isolated for ontology analyses. GO analyses were performed using Database for Annotation, Visualization and Integrated Discovery Bioinformatics Resource[62] and genes with terms relevant to axon growth (FDR < 0.05) based on literature review were user-selected for further analyses.

**Enhancer identification**. Enhancers relevant to pro-growth genes were identified by running the ABC algorithm[25] using code described here—https://github.com/broadinstitute/ABC-Enhancer-Gene-Prediction. ENCODE datasets used for the analysis are as follows: ATAC-Seq (ENCSR310MLB, ENCSR836PUC), H3K27Ac histone chromatin immunoprecipitation sequencing (ChIP-seq) (ENCSR094TTT, ENCSR428OEK), RNA-seq (ENCSR362AIZ, ENCSR080EVZ), and HiC (GSE96107). The algorithm has three sequential steps as follows: definition of candidate enhancers, quantification of enhancer activity, and calculation of ABC scores. For definition of candidate enhancers, indexed and sorted bam files of ATAC-Seq and H3K27Ac ChIP-seq datasets were supplied and peaks were called using MACS2, with the argument --nStrongestPeaks 15000. For quantification of enhancer activity, reads counts were calculated using peak files processed in the previous step, to yield a final list of candidate enhancer regions with ATAC-seq and

H3K27ac ChIP-seq read counts within gene bodies and promoters. Finally, ABC scores were calculated using arguments -hic_resolution 5000, -scale_hic_using_powerlaw, and a threshold of 0.01. The default threshold of 0.01 corresponds to ~70% recall and 60% precision. All enhancer–gene pairs that passed the above threshold were used for subsequent analyses.

**Co-occupancy motif analyses**. TF-binding site/motif analysis on pro-growth gene promoters and enhancers was performed using opossum v3.0 software[27]. For promoter analyses, the list of pro-growth genes was supplied along with promoter coordinates to be used for scanning (upstream/downstream of TSS—1000/300 bps). For enhancer motif analyses, a unified file containing a list of FASTA formatted sequences corresponding to pro-growth enhancer regions was supplied.

Mouse anchored TF-cluster analyses were performed using search parameters—JASPAR CORE profiles that scan all vertebrate profiles with a minimum specificity of 8 bits, conservation cutoff of 0.40, matrix score threshold of 90%, and results sorted by $Z$-score ≥ 10. TFs were sorted into three categories as follows: TFs with over-represented sites within pro-growth promoters, TFs with over-represented sites within pro-growth enhancers, and TFs with over-represented binding sites within both pro-growth promoters and enhancers.

**Network analyses**. Hit TF target gene networks were constructed using ENCODE ChIP-Seq data for hit TFs using the Harmonizome database[63]. For each of the 12 hit TFs, TF target gene lists were retrieved from Harmonizome individually and merged before visualization on the Cytoscape platform (v3.7.1). Cytoscape plug-ins ClueGO and CluePedia were used for network visualization. iRegulon plug-in on the Cytoscape platform was used to expand the network one level and bring in additional regulators. Next, we calculated connectivity scores to assign centrality using the Network Analyzer option within Cytoscape 3.7.1. TFs were ranked in descending order of connectivity scores to assign centrality. Core TFs were those TFs that had the highest connectivity scores and were centrally located in the network, followed by TFs with decreasing connectivity scores that occupied peripheral positions in the network. Upstream regulator analysis on Klf6 target genes was run on differentially expressed genes listed in ref. [7] using motif analyses described above. Motif analyses were done in batches such that each batch had genes belonging to one functional subnetwork to identify Klf6 synergizers by functional category.

**Cloning and virus production**. Constructs for candidate genes were purchased from Dharmacon or Origene, and relevant accession numbers for all 62 candidate TFs are summarized in Supplementary Data 2. mScarlet was a gift from Dorus Gadella (RRID:Addgene_85044), pAAV-CAG-tdTomato (codon diversified) was a gift from Edward Boyden (RRID:Addgene_59462), and mNeonGreen sequences were synthesized by Genscript, based on the amino acid sequence of ref. [64]. For viral production, genes were cloned into an AAV-CAG backbone (Addgene Plasmid #59462) using standard PCR amplification[12]. Maxipreps were prepared by Qiagen Endo-free kits and fully sequenced, and AAV9-tdTomato (Addgene Plasmid #59462) and AAV2-Retro of all other constructs were produced at the University of North Carolina Viral Vector Core and brought to $1 \times 10^{13}$ particles per ml in sterile saline prior to injection. Viruses were mixed at a 3 : 3 : 2 ratio of combinatorial test genes and tracer; viral treatments are detailed in Supplementary Data 3.

**Cortical cell culture and analysis of neurite outgrowth**. All animal procedures were approved by the Marquette University Institutional Animal Care and Use Committee. Cortical neurons were prepared from early postnatal (P5–P7) Sprague–Dawley rat pups (Harlan), with each experimental replicate derived from a separate litter. Procedures for dissociation, transfection, immunohistochemistry, imaging, and neurite outgrowth analysis were performed as in ref. [7]. Plasmid expressing nuclear-localized EGFP, mixed with text plasmids at a 1 : 3 ratio, served to identify transfected cells. Each 24-well culture plate included three standard treatments: enhanced blue fluorescent protein (EBFP) control, Klf6 plasmid mixed in equal parts with mCherry control, and Klf6 mixed in equal parts with VP16-Stat3. Remaining wells contained Klf6 mixed with candidate TFs. All morphological values were normalized to within-plate EBFP control values, the Klf6-only treatment served to set the level of Klf6's individual effect, and the Klf6/VP16-stat3 combination confirmed sensitivity of the assay plate to TF synergy[7]. For all cell culture experiments, neurite length from a minimum of 150 cells per treatment was averaged and each experiment was repeated a minimum of three times on separate days. These averaged values were the basis for ANOVA with Fisher's multiple comparisons. Neurite outgrowth values, number of cells analyzed for every screening experiment are summarized in Supplementary Data 2.

**Viral delivery to cortical neurons and pyramidotomy injuries**. In vivo experiments were performed in a double-blind manner, with non-involved lab personnel maintaining blinding keys. Experiments used adult C57BL/6 mice, 8–10 weeks of age at the start of the experiment. The first pyramidotomy experiment used female mice and the second pyramidotomy and spinal crush experiment used mixed sex at approximately equal numbers; sex had no significant effect on axon growth in either experiment ($p > 0.05$, two-way ANOVA). Animals were randomized prior to

viral treatment, with each surgical day including equal numbers from each group. Cortical neurons were transduced using intracerebral microinjection[7,8]. Briefly, mice were anesthetized with ketamine/xylazine (100/10 mg/kg, intraperitoneally), mounted in a stereotactic frame, and skull exposed and scraped away with a scalpel blade. Virus particles (0.5 μl) were delivered at two sites located 0 mm/1.3 mm and 0.5 mm/1.3 mm (anterior/lateral from Bregma) in pyramidotomy experiments, and −0.6 mm/1.3 mm and −1.2 mm/1.3 mm (anterior/lateral from Bregma) in spinal crush experiments, at a depth of 0.55 mm and at a rate of 0.05 μl/min using a pulled glass micropipette connected to a 10 μl Hamilton syringe driven by a programmable pump (Stoelting QSI), with 1 min dwell time. For spinal injections, mice were mounted in a custom spine stabilizer and viral particles or Cholera Toxin Subunit B conjugated to Alexa Fluor 647 (CTB-647) in sterile 0.9% NaCl (C22841-Thermofisher, Waltham, MA, final concentration of 2%) were injected to the spinal cord through a pulled glass micropipette fitted to a 10 μl Hamilton syringe driven by a Stoelting QSI pump (catalog # 53311) (pumping rate: 0.04 μl/min) between C4 and C5 vertebrae, 0.35 mm lateral to the midline, and to depths of 0.6 mm and 0.8 mm, 0.5 μl at each depth. Unilateral pyramidotomy was performed[8,12]. Briefly, a ventral midline incision was made to expose the occipital bone, the ventrocaudal part of which was removed using fine rongeurs. The dura was punctured and the right pyramid cut completely using a micro feather scalpel. For spinal crush injuries, mice were anesthetized and mounted in a spine stabilization device, a laminectomy performed at T12 vertebrae/T10/11 spinal cord, and forceps of width 250 μm used to crush the cord for 10 s, then grip-reversed and repeated for another 10 s.

**RNAScope**. RNAscope kits are commercially available from Advanced Cell Diagnostics (ACD). We utilized the Multiplex v2 system for all RNAScope experiments (Cat-323100, ACD). Mouse brains were snap-frozen on dry ice and cryosectioned (25 μm). Slides were baked at 60 °C for 40 min followed by RNAScope procedures according to manufacturer's instructions. Briefly, hydrogen peroxide solution was applied to baked slices and incubated for 10 min at room temperature (RT). Next, slices were incubated in boiling antigen retrieval solution (<98 °C) for 5 min. Following retrieval, slices were washed in Nuclease-free water three times, 5 min per wash. Then, brain slices were dehydrated in 100% ethanol briefly followed by treatment with ProteaseIII for 20 min at 40 °C. Following protease incubation, slices were washed in Nuclease-free water three times, 5 min per wash, and probes were applied and allowed to incubate for 2 h at 40 °C. The following probes were purchased off the catalog RNAscope® Probe-Mm-Klf6 (Cat-426901, ACD), RNAscope® Probe-Mm-Nkx3-2 (Cat- 526401, ACD), and RNAscope® Probe-Mm-Rarb (Cat- 463101, ACD). We designed a custom probe for Nr5a2, targeting a region common to all variants (RNAscope® Probe-Mm-Nr5a2-O1-C2, Cat- 547841-C2, ACD). All probes were detected with tyramide signal amplification plus fluorophore used at 1 : 750 dilution. Before mounting the slices, a brief 5 min incubation with 4′,6-diamidino-2-phenylindole was performed to label the nuclei. All RNAScope experiments were carried out between 1 and 3 days post cryosectioning, to ensure tissue integrity.

**Immunohistochemistry**. Adult animals were perfused with 4% paraformaldehyde (PFA) in 1× phosphate-buffered saline (PBS) (15710-Electron Microscopy Sciences, Hatfield, PA), brains, and spinal cords removed, and post-fixed overnight in 4% PFA. Transverse sections of the spinal cord or cortex were embedded in 12% gelatin in 1× PBS (G2500-Sigma Aldrich, St.Louis, MO) and cut via Vibratome to yield 100 μm sections. Sections were incubated overnight with primary antibodies PKCγ (SC C-19, Santa Cruz, Dallas, TX, 1 : 500, RRID: AB_632234), GFAP (DAKO, Z0334 1 : 500, RRID:AB_10013482), or Cd11b (Invitrogen 14-01120-82 1 : 500, RRID:AB_2536484), or Eomes (ab23345, RRID:AB_778267, 1 : 500), and rinsed and then incubated for 2 h with appropriate Alexa Fluor-conjugated secondary antibodies (R37116/7, Thermofisher, Waltham, MA, 1 : 500). For TUNEL staining (In Situ Cell Death Kit, Roche), fresh-frozen transverse cryostat sections (30 μm) were post-fixed in 4% PFA 15 min, incubated in ethanol/acetic acid for 10 min, 0.4% Triton for 30 min, and with probe mixture for 1 h[8]. Staurosporine control (1 μl, 1 mm; Sigma) was cortically injected 2 days prior to killing. Fluorescent images were acquired using Olympus IX81 or Zeiss 880LSM microscopes.

**Quantification of axon growth**. In pyramidotomy experiments, axon growth was quantified from four 100 μm transverse sections of the spinal cord of each animal that spanned C2 to C6 spinal cord. Starting from a complete series of transverse sections, the rostral-most was selected from C2 as identified by the morphology of the ventral horns of the spinal cord, and the next three selected at 1.6 mm intervals in the caudal direction. Each section was imaged on Nikon confocal microscope (Nikon AR1+) using a ×20 Plan Apochromatic (MRD00205, NA 0,75) objective, gathering 20 μm of images at 5 μm intervals and then creating a maximum intensity projection. On the resulting image, virtual lines of 10 μm width were placed at 200, 400, and 600 μm from the midline and intersection of tdTomato+ axons with these lines were quantified. For normalization, 100 μm transverse sections of the medullary pyramids were prepared by vibratome sectioning and imaged on a Nikon confocal microscope (Nikon AR1+) using a ×60 Apochromatic Oil DIC N2 (MRD71600, NA 1.4) objective. Seven virtual lines of 10 μm width were distributed evenly across the medial/lateral axis of the medullary pyramid, the

number of tdTomato+ axons visible in each was determined, and the total number of axons extrapolated based on the area of the entire pyramid and the area of the sampled regions. Fiber index was calculated as the average number of axons at each distance from the midline across the four replicate spinal sections, divided by the calculated number of axons detected in the pyramid. Counting of digital images was performed by three blinded observers, with final values reflecting the average. Exclusion criteria for pyramidotomy experiments were animals with <80% decrease in PKCγ in the affected CST. For spinal crush injuries, quantification and normalization of cross-midline sprouting was nearly identical to those for pyramidotomy, with the exception that quantification was performed in four horizontal sections of the spinal cord, with the sampling area set 400 μm from the midline. To quantify axon branching, the following procedure was employed. First, in transverse sections of cervical spinal cord, tdTomato+ CST axons were identified in a sampling region between 200 and 600 μm from the midline. Next, axon segments traced, with the requirement that sampled segments must remain in a single confocal imaging plane at ×60 magnification for a minimum of 100 μm traced length. Finally, using Z-stacks to definitively distinguish true branches from near-plane intersections, the number of branches was counted in each sampled segment. A minimum of 5 mm of total length, from three separate spinal sections, were sampled for each animal.

**Fluorescence-activated nuclei sorting**. Equal numbers of male and female mice were used in every experiment in concordance with National Institutes of Health guidelines. Adult mice received retrograde injections of viral vectors for Ctrl treatment, Klf6-alone treatment, or combined Klf6 + Nr5a2 treatment. One week later, animals were challenged with pyramidotomy injuries, as described above. One week post injury, animals were killed and the motor cortices were dissected. Tissue was collected only from the hemisphere contralateral to the pyramidotomy injury, thus isolating spared neurons and not neurons that experienced direct axotomy. Dissected cortices were minced finely using razor blades and transferred to pre-chilled 15 ml Dounce homogenizer filled with 3 ml Nuclear release buffer (320 mM Sucrose, 5 mM Cacl₂, 3 mM MgCl₂, 10 mM Tris-HCl, 0.3% Igepal). Tissue was dounced 15× while on ice and filtered sequentially via a 50 and 20 μm filter, and used as input for flow cytometry. Dissociated nuclei were flow-sorted on a BD FACS Melody using an 80 μm nozzle and a sequential gating strategy (Sort type: Purity). Specifically, nuclei were first gated by side scatter height (SSC-H) vs. forward scatter height (FSC-H) and side scatter area SSC-A vs. FSC-A to separate debris vs. intact nuclei. Nuclei that passed these filters were then filtered by SSC-W vs. SSC-H, to eliminate potential doublets. Finally, nuclei were gated by levels of fluorescence marker such that only the brightest nuclei are collected to a goal of ~30,000 events for RNA-Seq and 4000 events for single-nuclei RNA-Seq.

**RNA-Seq data generation and analysis**. Following fluorescence-activated nuclei sorting, nuclei were sorted directly into Trizol, lysed, and RNA was extracted according to the manufacturer's instructions. Only samples with an RNA integrity number score of 8 and above were used for library prep. Total RNA was used for library prep using Takara SMARTer stranded total RNA-Seq kit v2 according to the manufacturer's instructions. Samples were sequenced at the University of Wisconsin-Madison Genomics core to a depth of 25 million paired-end reads on an Illumina HiSeq platform, with two replicates per treatment. Raw FASTQ files were fed into ENCODE consortia RNA-Seq pipeline described here— https://github.com/ENCODE-DCC/rna-seq-pipeline. Read alignment was performed using STAR[65] and transcript quantification performed using Kallisto[66]. Transcript counts from Kallisto were used for performing differential gene expression analyses using EdgeR[36]. Transcripts were considered significant if they met the statistical requirement of having a corrected $p$-value of <0.05, FDR < 0.05. Differentially expressed genes were further filtered to only include transcripts with FPKM > 5 and log2Fold change > 1. RNA-Seq datasets have been deposited with NCBI GEO (PRJNA630017). Network analysis on significantly differentially expressed genes (upregulated) was performed on the cytoscape platform v3.7.1 using plug-ins clueGo and CluePedia. Functional analysis mode was selected and the following GO categories were selected for analyses—GO Biological Process, Kyoto Encyclopedia of Genes and Genomes pathways, Reactome Pathways, and Wiki pathways. Network specificity was set to medium and GO tree interval was set between 3 and 8. GO term κ-score was set to default values (0.4) and leading group term was set to rank by highest significance. Lastly, GO term fusion was also selected and only subnetworks with significantly enriched GO terms were used to generate network visualizations ($p < 0.05$, right-sided hypergeometric test with Bonferroni correction, Cytoscape). Expression heatmaps were assembled using heatmapper.ca, using average linkage and Euclidean distance parameters for clustering.

**Single-nuclei RNA-Seq data generation and analysis**. Equal numbers of male and female mice were used in every experiment in concordance with NIH guidelines. Adult mice (8 weeks) received retrograde injections of viral vectors for Ctrl treatment or combined Klf6 + Nr5a2 treatment. One week later, animals were challenged with pyramidotomy injuries, as described above. One week post injury, animals were killed and the motor cortices were dissected. Dissected cortices were frozen on dry ice and stored at −80° until library preparation. Frozen tissue was transferred to a pre-chilled 3 ml Kimble dounce homogenizer filled with 4 ml Nuclei

Lysis Buffer (Sigma, NUC101), supplemented with RNAase inhibitors (10 mg/ml) (RNAase OUT, Thermofisher 10777019). Tissue was homogenized (20 strokes with pestle A and 25 stroked with pestle B) and allowed to incubate on ice for 5 min. Following incubation, nuclei was centrifuged for 5 min at 4° (800 G, low brake and acceleration) and the pellet was resuspended in 4 ml Nuclei lysis buffer. Homogenate was incubated for 5 min on ice. Following incubation, nuclei was centrifuged as described above and resuspended in 0.5 ml 1× PBS with 1% bovine serum albumin. Dissociated nuclei were filtered using a 20 μm filter and flow-sorted on a BD FACS Melody using an 80 μm nozzle to a goal of ~4500–5000 events (gate strategy described above). Nuclei were sorted directly into 10× RT buffer (enzyme added only prior to Gel Bead-in Emulsion (GEM) generation) and library preparation was performed on the 10× chromium platform according to manufacturer's instructions (10× Genomics- Next GEM, Cat # PN1000121). Samples were sequenced at the University of Wisconsin-Madison Genomics core to a depth of ~70,000 reads/nuclei (3000–4000 nuclei per library) on an Illumina NovaSeq platform. Raw FASTQ files were fed into the Cellranger pipeline (default parameter) described here—https://github.com/10XGenomics/cellranger. Datasets were then integrated using SEURATv3[39] and differential expression testing was performed according to default conditions (non-parametric Wilcoxon rank-sum test).

**Reporting summary**. Further information on experimental design is available in the Nature Research Reporting Summary linked to this paper.

## Data availability

There are no restrictions on data availability. All the datasets generated and analyzed during the current study are available in the NCBI repository (PRJNA630017). The following publicly available datasets were also analyzed in this study-ATAC-Seq (ENCSR310MLB, ENCSR836PUC), H3K27Ac histone ChIP-seq (ENCSR094TTT; https://www.encodeproject.org/files/ENCFF195BGJ/, ENCSR428OEK), RNA-seq (ENCSR362AIZ, ENCSR080EVZ), and HiC (GSE96107). Source data are provided with this paper.

## Code availability

There was no custom code development and all software used in data analyses are previously published, open access, and have been cited under the relevant methods section. Links to relevant software repositories/documentation is listed here. WGCNA— https://horvath.genetics.ucla.edu/html/CoexpressionNetwork/Rpackages/WGCNA/; RNA-Seq analyses—https://github.com/ENCODE-DCC/rna-seq-pipeline; EdgeR— https://www.bioconductor.org/packages/release/bioc/html/edgeR.html; Activity-by-contact (ABC)—https://github.com/broadinstitute/ABC-Enhancer-Gene-Prediction; oPOSSUM 3.0—http://opossum.cisreg.ca/oPOSSUM3/; Cytoscape-ClueGO—http://www.ici.upmc.fr/cluego/cluegoDocumentation.shtml; Harmonizome—https://amp.pharm.mssm.edu/Harmonizome/; iRegulon—http://iregulon.aertslab.org/; Cellranger— https://github.com/10XGenomics/cellranger. SEURAT—https://github.com/satijalab/seurat; FACSChorus—https://www.bdbiosciences.com/en-us/instruments/research-instruments/research-software/flow-cytometry-acquisition/facschorus-software; and DAVID—https://david.ncifcrf.gov/summary.jsp.

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

# ARTICLE

40. Wang, Z., Winsor, K., Nienhaus, C., Hess, E. & Blackmore, M. G. Combined chondroitinase and KLF7 expression reduce net retraction of sensory and CST axons from sites of spinal injury. *Neurobiol. Dis.* **99**, 24–35 (2017).

41. Robinson-Rechavi, M., Escriva Garcia, H. & Laudet, V. The nuclear receptor superfamily. *J. Cell Sci.* **116**, 585–586 (2003).

42. Yip, P. K. et al. Lentiviral vector expressing retinoic acid receptor β2 promotes recovery of function after corticospinal tract injury in the adult rat spinal cord. *Hum. Mol. Genet.* **15**, 3107–3118 (2006).

43. Hale, M. A. et al. The nuclear hormone receptor family member NR5A2 controls aspects of multipotent progenitor cell formation and acinar differentiation during pancreatic organogenesis. *Development* **141**, 3123–3133 (2014).

44. Stergiopoulos, A. & Politis, P. K. Nuclear receptor NR5A2 controls neural stem cell fate decisions during development. *Nat. Commun.* **7**, 12230 (2016).

45. Liu, Y. et al. A sensitized IGF1 treatment restores corticospinal axon-dependent functions. *Neuron* **95**, 817–833.e4 (2017).

46. Chandran, V. et al. A systems-level analysis of the peripheral nerve intrinsic axonal growth program. *Neuron* **89**, 956–970 (2016).

47. Puttagunta, R. et al. PCAF-dependent epigenetic changes promote axonal regeneration in the central nervous system. *Nat. Commun.* **5**, 3527 (2014).

48. Du, K. et al. Pten deletion promotes regrowth of corticospinal tract axons 1 year after spinal cord injury. *J. Neurosci.* **35**, 9754–9763 (2015).

49. Leibinger, M. et al. Transneuronal delivery of hyper-interleukin-6 enables functional recovery after severe spinal cord injury in mice. *Nat. Commun.* **12**, 391 (2021).

50. Kadoya, K. et al. Spinal cord reconstitution with homologous neural grafts enables robust corticospinal regeneration. *Nat. Med.* **22**, 479–487 (2016).

51. Tuszynski, M. H. & Steward, O. Concepts and methods for the study of axonal regeneration in the CNS. *Neuron* **74**, 777–791 (2012).

52. Chen, M. & Zheng, B. Axon plasticity in the mammalian central nervous system after injury. *Trends Neurosci.* **37**, 583–593 (2014).

53. Ewan, E. E., Carlin, D., Goncalves, T. M., Zhao, G. & Cavalli, V. Ascending dorsal column sensory neurons respond to spinal cord injury and downregulate genes related to lipid metabolism. *Sci. Rep.* https://doi.org/10.1038/s41598-020-79624-0 (2021).

54. Haffner, M. C., De Marzo, A. M., Meeker, A. K., Nelson, W. G. & Yegnasubramanian, S. Transcription-induced DNA double strand breaks: both oncogenic force and potential therapeutic target? *Clin. Cancer Res.* **17**, 3858–3864 (2011).

55. Sebastian, R. & Oberdoerffer, P. Transcription-associated events affecting genomic integrity. *Philos. Trans. R. Soc. Lond. B Biol. Sci.* **372**, 20160288 (2017).

56. Onishi, K. et al. Genome stability by DNA polymerase β in neural progenitors contributes to neuronal differentiation in cortical development. *J. Neurosci.* **37**, 8444–8458 (2017).

57. Baleriola, J. et al. Increased neuronal death and disturbed axonal growth in the Polμ-deficient mouse embryonic retina. *Sci. Rep.* **6**, 25928 (2016).

58. Krishnan, A. et al. A BRCA1-dependent DNA damage response in the regenerating adult peripheral nerve milieu. *Mol. Neurobiol.* **55**, 4051–4067 (2018).

59. Kim, D., Langmead, B. & Salzberg, S. L. HISAT: a fast spliced aligner with low memory requirements. *Nat. Methods* **12**, 357–360 (2015).

60. Pertea, M., Kim, D., Pertea, G. M., Leek, J. T. & Salzberg, S. L. Transcript-level expression analysis of RNA-seq experiments with HISAT, StringTie and Ballgown. *Nat. Protoc.* **11**, 1650–1667 (2016).

61. Langfelder, P. & Horvath, S. WGCNA: an R package for weighted correlation network analysis. *BMC Bioinformatics* **9**, 559 (2008).

62. Huang, D. W., Sherman, B. T. & Lempicki, R. A. Systematic and integrative analysis of large gene lists using DAVID bioinformatics resources. *Nat. Protoc.* **4**, 44–57 (2009).

63. Rouillard, A. D. et al. The harmonizome: a collection of processed datasets gathered to serve and mine knowledge about genes and proteins. *Database* **2016**, baw100 (2016).

64. Shaner, N. C. et al. A bright monomeric green fluorescent protein derived from Branchiostoma lanceolatum. *Nat. Methods* **10**, 407–409 (2013).

65. Dobin, A. et al. STAR: ultrafast universal RNA-seq aligner. *Bioinformatics* **29**, 15–21 (2013).

66. Bray, N. L., Pimentel, H., Melsted, P. & Pachter, L. Near-optimal probabilistic RNA-seq quantification. *Nat. Biotechnol.* **34**, 525–527 (2016).

## Acknowledgements

This work was supported by grants from NINDS (5R01NS083983 and R21NS106309), The Craig Neilsen Foundation, The Bryon Riesch Paralysis Foundation, and computational allocations from NSF-XSEDE. We thank Erik Van Newenhizen for technical assistance. We acknowledge ENCODE consortia for generating the developmental time-series NGS datasets used in this study. We acknowledge the use of Biorender (academic license) to generate illustrations and graphics used in the manuscript.

## Author contributions

M.G.B. and I.V. conceived and planned the experiments. I.V., V.M., Z.W., M.T.S., E.E., A.C., Z.B., D.G., M.C., and G.O. carried out the experiments. M.B.M. and I.V. contributed to the interpretation of the results. M.B.M. and I.V. were involved in writing the manuscript.

## Competing interests

The authors declare no competing interests.
