## [Peer Review File · Nature Communications]

Reviewers' comments:

Reviewer #1 (Remarks to the Author):

Transcription factors (TFs) are powerful in regulating neuronal function in normal and disease conditions. To date, most studies on CNS regeneration have focused on identifying single TF of which ectopic expression in mature neurons may lead to improved axonal growth. Since multiple TFs may act synergistically, identifying an optimal set of TFs may exert a greater effect on axonal growth than when a single TF is used. The most significant finding of this paper is that the authors have provided the first and convincing evidence that combined expression of two TFs, i.e., Klf6 and Nr5a2, synergistically enhanced spared corticospinal tract (CST) axonal growth across the midline of the spinal cord following a unilateral pyramidotomy, as compared to each of the individual factors.

Another significance of this study is that the authors developed a systematic pipeline aimed at discovering growth-relevant TF combinations centered on TF co-occupancy of regulatory DNA, and used it to predict factors that potentiate the effects of proregenerative Klf6.

Through high content screening of neurite outgrowth, they identified 12 candidates that showed cooperative activity. Among them, combined Klf6 and Nr5a2 showed the strongest effect on axonal growth in vivo. Thus, using combinatorial approaches including interlocking bioinformatics, high content screening, and in vivo testing, the authors identified new TF combinations that synergize to enhance CST axonal growth.

In general, the study has been rigorously conducted and is of high quality. The experiments were well designed and properly controlled. The findings are important and should be of interest to a broad audience interested in CNS injury and repair. The systematic pipeline approach may shed light on identifying new combinations of TFs for a broader application. I have only a few additional comments that need to be addressed, as described below:

1. Fig 1. Is too busy to be appreciated. Some font sizes are too small and difficult to read.

The figure may be rearranged to create space for enlarging the font size.

2. In Fig. 2, the image quality is poor. In the black and white images, individual axonal fibers across the midline were not clearly shown. I suggest the authors include high power images to show clearly the individual axonal growth. In "h" and "l", both images are labeled "Klf6 + Nkx3.2", but their growth patterns across the midline are different. Please double check.

3. It is not known whether the CST axons, after crossing, make connections with neurons on the contralateral side and, if so, such circuitry reorganization will bring about any meaningful functional changes. This could be something that the authors could consider in their future studies.

4. Likewise, unilateral pyramidotomy could only test sprouting of spared CST axons, it would be interesting to see if this combined TF strategy could be applied to promote true regeneration of cut axons and functional outcomes in a spinal cord injury mode.

5. Will cortical injection of AAV cause local inflammation, glial response, and cell death? Did the authors check histopathology at the cortical injection site?

6. It would be helpful to include "n" number in figure legends wherever appropriate.

7. In Fig. 4, the font sizes are too small to be appreciated. Please modify.

8. In the methods section, the strain, sex, and age of mice used in this study should be indicated.

Xiao-Ming Xu

Reviewer #2 (Remarks to the Author):

In this paper, the authors suggest a bioinformatic and screening platform, focusing on the concept of TF co-occupancy and propose a synergy between the transcription factors Klf6 and Nr5a2 that leads to improved axon growth after CNS injury. The authors first show using an in vitro based screen that a set of transcription factors synergize with Klf6 to promote axon growth in cultured cortical neurons. They then move to a pyramidotomy assay to test which transcription factor combined with Klf6 enhances sprouting. They identify Nr5a2 as their top hit. Finally, they perform RNAseq of sorted cortical neurons to unravel the gene modules triggered by klf6/Nr5a2 co-expression. This manuscript is similar conceptually to what they previously reported (Wang et al 2018, Scientific Reports, 2), in which they identified synergy between Klf6 and STAT3 in promoting axon growth in cultured CNS neurons. This current study moves the concept of TF synergy to in vivo models and is thus relevant to the broad field of axon regeneration. There are however a number of issues that need to be addressed to improve the clarity, framing and relevance of the manuscript.

Major points

1. The rationale for the choice of CNS injury needs to be better explained. The authors have previously shown that forced KLF6 expression increases axon regeneration by directly injured CST neurons. Here they opted for a pyramidotomy model, in which KLF6 combined with Nr5a2 increases CST axon sprouting from spared neurons. It would be important to the field to know if KLF6 combined with Nr5a2 can also enhance re-growth of damaged neurons.
2. If the authors know that STAT3 synergizes with klf6, why was this combination not tested in the in vivo assay?
3. A lot of information is missing in regards to the experimental rationale and procedures making it hard to understand the flow of the study. This is the case for experiments describing the phenotypic screen in Figure 1. Stat3 is expressed with a vp16 domain. Is that the case for all other TFs tested in that assay? If not, why was vp16 not included? Representative images of culture neurons should be shown and more details on quantification methods should be presented.
4. The experimental design in Figure 4 is also unclear and confusing. Nuclei are sorted for RNAseq analysis to probe the underlying mechanisms of growth promotion. According to the diagram the injured neurons in Figure 4a are collected. But in all other experiments, the effect of klf6 and synergizing TF is assessed on the non-injured sprouting neurons. Does the injury in itself change gene expression in CSMN? How does the analysis of gene expression in injured CSMN relate to the observed sprouting effect from non-injured neurons? If KLF6 combined with Nr5a2 were shown to also enhance re-growth of damaged neurons (see point #1 above), then this RNAseq experiment would be more relevant.
5. The similarity in gene expression changes provided by klf6 expression in vitro (from their previous study Wang et al 2018) and the current in vivo data set is very interesting and warrants more discussion. This similarity highlights the usefulness of in vitro approaches.
6. A clear evidence for co-expression of the two transcription factors in the same neuron together with the tracer is necessary, especially because this is the main claim of this study. The presented in situ images in Supp figure 4 are not convincing as presented. It is not clear in which cell type the signal is detected, as there is no co-staining with a neuronal marker. Quantification of the percentage of cells expressing both transcripts should be shown. How were the RNA scope probe designed to detect specifically the expressed but not endogenous TF?

Minor points

- Supp figure 1b should be presented in main figure, as this is important to understand what is measured.
- There are no references in the text for fig.2b,c
- Figure2- h and I both have images describing co-expression of Klf6+Nkx3.2, but the effects are different
- Venn diagrams of DE genes in the RNAseq related to Figure 4 would be helpful to understand the differences between single and double TF expression.
- Table supplemental 3 - There are some mix-ups in the virus descriptions.
- Add explanation why for the TF expression AAV2 was used and for the TdTomato expression AAV9 was used - is there a difference in infection efficiency or cell specificity of expression?
- Overall a better definition is needed when using the terms axonal growth, extension, sprouting and regeneration.
- There are no details on FANS in the method section.
- Figure 4g-I Regulatory network analysis of genes upregulated after combined Klf6 and Nr5a2. Is that all the genes regulated by klf6+Nr5a2 or only the 192 that were not change by either TF alone.

Reviewer #3 (Remarks to the Author):

In their manuscript, Venkatesh and colleagues present a pipeline for the identification of transcription factors (TFs) that could promote axon growth and regeneration, especially when provided together. To identify such potential synergy, the authors use Klf6, known to promote neurite growth following ectopic expression, as a starting point. First, computational methods are used to identify other TFs that co-occupy regulatory DNA with Klf6. This is followed by a high-throughput screen to explore which of these TFs actually promotes axon growth from cortical neurons when co-transfected with Klf6. The outcome of this screen is subsequently used to perform network analyses that identify the 'core' TFs Nr5a2, Eomes and Rarb. AAV-mediated overexpression of these TFs in the cortex following CST injury is then induced and shown to facilitate enhanced regeneration, especially when combinations of TFs are provided. Finally, neurons are collected from the transduced cortex and subjected to RNAseq to begin to provide insight into underlying molecular mechanisms. In all, this is a nice study that emphasizes the strength of combining TFs for enhancing axon (re)growth and that provides a pipeline for the identification of the most powerful candidates.

I have a few points that need to be addressed prior publication:

- The in vitro screen presented in Fig. 1b lacks morphological details. No transfected neurons are shown and it is hard to assess the quality/robustness of this screen. Are there other effects, e.g. on cell death, axon branching etc. The presentation of these data needs to be expanded, perhaps as part of the Suppl. Data. Are there any quality controls for the screen?

- In line 157-164, data are shown that identify TFs related to Klf6 (by analysis of neurons overexpressing Klf6) that are also found in the previous computational approach. It makes me wonder which is the added benefit of the analysis as shown in Fig. 1. Wouldn't the identification of TFs following overexpression of a specific TF (such as Klf6) be a more straightforward approach?

- The focus of this manuscript is very much on axon growth but the in vivo effects presented could also be a result (in part) of enhanced branching, increased survival, reduced sensitivity to an inhibitory environment etc. At least some experiments need to be performed to control for this.

- Fig. 3e: Why is there no effect of NR5A2+RARβ at 200 micron while there is at 400 and 600 micron?

- Fig. 4: This part is very open-ended. At least some of the regulated genes should be confirmed in vivo and for this type of journal one wonders whether functionally counteracting a downstream gene in vitro or in vivo following TF overexpression is not a required minimum.

We thank the reviewers for their helpful and insightful input, which inspired a series of additional experiments and re-analyses. We hope you will agree that these new data both strengthen and enrich the manuscript. We also appreciate being alerted to errors and points of confusion, which we have made every effort to correct. We detail our response to each comment point-by-point below.

REVIEWER COMMENTS

Reviewer #1 (Remarks to the Author):

Transcription factors (TFs) are powerful in regulating neuronal function in normal and disease conditions. To date, most studies on CNS regeneration have focused on identifying single TF of which ectopic expression in mature neurons may lead to improved axonal growth. Since multiple TFs may act synergistically, identifying an optimal set of TFs may exert a greater effect on axonal growth than when a single TF is used. The most significant finding of this paper is that the authors have provided the first and convincing evidence that combined expression of two TFs, i.e., Klf6 and Nr5a2, synergistically enhanced spared corticospinal tract (CST) axonal growth across the midline of the spinal cord following a unilateral pyramidotomy, as compared to each of the individual factors.

Another significance of this study is that the authors developed a systematic pipeline aimed at discovering growth-relevant TF combinations centered on TF co-occupancy of regulatory DNA, and used it to predict factors that potentiate the effects of proregenerative Klf6. Through high content screening of neurite outgrowth, they identified 12 candidates that showed cooperative activity. Among them, combined Klf6 and Nr5a2 showed the strongest effect on axonal growth in vivo. Thus, using combinatorial approaches including interlocking bioinformatics, high content screening, and in vivo testing, the authors identified new TF combinations that synergize to enhance CST axonal growth.

In general, the study has been rigorously conducted and is of high quality. The experiments were well designed and properly controlled. The findings are important and should be of interest to a broad audience interested in CNS injury and repair. The systematic pipeline approach may shed light on identifying new combinations of TFs for a broader application. I have only a few additional comments that need to be addressed, as described below:

1. Fig 1. Is too busy to be appreciated. Some font sizes are too small and difficult to read. The figure may be rearranged to create space for enlarging the font size.

In response to this input, and similar input from other reviewers, we have now divided the original Figure 1. The revised **Figure 1** focuses only on the bioinformatic pipeline for TF discovery, making room for expanded font size. **Figure 2** is now devoted to the screening results and network analyses of hit genes, and similarly benefits from the additional space. We thank the reviewer for this helpful comment.

2. In Fig. 2, the image quality is poor. In the black and white images, individual axonal fibers across the midline were not clearly shown. I suggest the authors include high power images to show clearly the individual axonal growth. In “h” and “l”, both images are labeled “Klf6 + Nkx3.2”, but their growth patterns across the midline are different. Please double check.

We apologize for the poor image quality in the original pdf, and have improved the size and resolution of the representative images (now **Figure 3**). We have also corrected the error in labeling.

3. It is not known whether the CST axons, after crossing, make connections with neurons on the contralateral side and, if so, such circuitry reorganization will bring about any meaningful functional changes. This could be something that the authors could consider in their future studies.

We agree that this a very important future direction, and we have added mention of this to the revised discussion.

4. Likewise, unilateral pyramidotomy could only test sprouting of spared CST axons, it would be interesting to see if this combined TF strategy could be applied to promote true regeneration of cut axons and functional outcomes in a spinal cord injury mode.

To respond to this request (also made by reviewer 3) we performed a spinal injury experiment (**Figure 6**) in which Klf6/Nr5a2-treated axons were confronted with a spinal crush injury. In control animals and Klf6/Nr5a2-treated animals alike, CST axons formed clear retraction bulbs and failed to traverse the injury. From this we conclude that Klf6/Nr5a2 treatment is insufficient to confer regenerative ability in the face of the strongly inhibitory lesion environment. We did, however, detect an elevation of collateral sprouting by injured axons just rostral to the injury, which even extended across the midline and into contralateral spinal tissue. These data indicate that Klf6/Nr5a2 can act on directly injured neurons to increase growth, which manifests as collateral extension when axons are confronted with a severe crush injury. This sets the stage for future work in the field to explore the potential for Klf6/Nr5a2-stimulated axons to form relay circuits after partial spinal injuries, or to combine Klf6/Nr5a2 treatment with growth-permissive tissue grafts. We appreciate the reviewer’s input, which has strengthened the manuscript by expanding the work from only a pure sprouting model to a test of growth in directly injured axons.

5. Will cortical injection of AAV cause local inflammation, glial response, and cell death? Did the authors check histopathology at the cortical injection site?

To address this question we performed two additional experiments to examine the histopathology of the injection site. Both experiments examined GFAP, CD11B, and TUNEL at the site of viral delivery. In the first we examined the injection site just three days after injury, and in the second

we looked for longer-term effects specifically on CST neurons that were identified by retrograde labeling. These data are now presented in **Supplementary Fig. 7**. Some gliosis and inflammation were detected near the injection site, as expected, but qualitatively did not seem to differ between control and Klf6/Nr5a2 treated animals. Importantly, quantification of TUNEL-positive cells in the vicinity of the injection detected no difference between control and Klf6/Nr5a2-treated animals at either time point. We appreciate the reviewer raising this point, and believe that inclusion of these new data help to argue against indirect effects of Klf6/Nr5a2 on gliosis, inflammation, or altered cell death as drivers of the growth response.

6. It would be helpful to include “n” number in figure legends wherever appropriate.

We have checked the figure legend throughout and added n values.

7. In Fig. 4, the font sizes are too small to be appreciated. Please modify.

We apologize for the small font sizes and have fixed this in the new **Figure 7** (original figure 4).

8. In the methods section, the strain, sex, and age of mice used in this study should be indicated.

This information has been added, thank you.

Xiao-Ming Xu

Reviewer #2 (Remarks to the Author):

In this paper, the authors suggest a bioinformatic and screening platform, focusing on the concept of TF co-occupancy and propose a synergy between the transcription factors Klf6 and Nr5a2 that leads to improved axon growth after CNS injury. The authors first show using an in vitro based screen that a set of transcription factors synergize with Klf6 to promote axon growth in cultured cortical neurons. They then move to a pyramidotomy assay to test which transcription factor combined with Klf6 enhances sprouting. They identify Nr5a2 as their top hit. Finally, they perform RNAseq of sorted cortical neurons to unravel the gene modules triggered by klf6/Nr5a2 co-expression. This manuscript is similar conceptually to what they previously reported (Wang et al 2018, Scientific Reports, 2), in which they identified synergy between Klf6 and STAT3 in promoting axon growth in cultured CNS neurons. This current study moves the concept of TF synergy to in vivo models and is thus relevant to the broad field of axon regeneration. There are however a number of issues that need to be addressed to improve the clarity, framing and relevance of the manuscript.

Major points

1. The rationale for the choice of CNS injury needs to be better explained. The authors have previously shown that forced Klf6 expression increases axon regeneration by directly injured CST neurons. Here they opted for a pyramidotomy model, in which Klf6 combined with Nr5a2 increases CST axon sprouting from spared neurons. It would be important to the field to know if Klf6 combined with Nr5a2 can also enhance re-growth of damaged neurons.

Following this suggestion (also made by Reviewer 1), we conducted a new experiment in which combined Klf6 and Nr5A2 were expressed in CST neurons, which were confronted with a crush injury in thoracic spinal cord. These new data are reported in new **Figure 6**. Klf6/Nr5a2 did not produce growth into the lesion site, and also did not affect retraction distance. We did however detect a significant increase in growth by collateral branches formed just above the injury, which were most apparent when they extended across the midline and into contralateral spinal cord. We conclude that Klf6/Nr5a2 expression is insufficient to overcome inhibitory cues expressed in the injury, but does act in directly injured axons to increase collateral sprouting. This characteristic may be quite useful in situations of partial spinal injury or in combination with growth-permissive tissue grafts, for example in fostering potential relay circuits. We speculate on this possibility in the revised discussion. We appreciate the encouragement to move ahead with this key test, and agree that the field benefits by inclusion of the new results.

2. If the authors know that STAT3 synergizes with klf6, why was this combination not tested in the in vivo assay?

STAT3 was previously identified as a potential synergizer with Klf6, and as the reviewer may guess has been the subject of ongoing *in vivo* experiments in a separate project in the lab. The data so far have proven somewhat inconsistent, and additional work will be needed to definitively rule in or out an interactive effect. In this manuscript we focus on novel interactions between Klf6 and other TFs. Although we appreciate the reviewer's interest in interactive effects between Klf6 and Stat3 *in vivo*, we would argue that the novel finding of strong synergy between Klf6 and Nr5a2, with supporting molecular mechanisms, stands as an independent and interesting finding.

3. A lot of information is missing in regards to the experimental rationale and procedures making it hard to understand the flow of the study. This is the case for experiments describing the phenotypic screen in Figure 1. Stat3 is expressed with a vp16 domain. Is that the case for all other TFs tested in that assay? If not, why was vp16 not included? Representative images of culture neurons should be shown and more details on quantification methods should be presented.

We have expanded both Methods and Results to include additional details to screening procedures, with a focus on the quality control measure incorporated into each screening plate. In addition, as requested we now include representative images (**Fig. 2b**). We have also greatly increased the data

available from the screen by adding **Supplementary Fig. 1**, which shows effects on additional morphological parameters (e.g. branching, growth with and without branches included, etc). We hope that this additional information helps clarify and enrich this aspect of the manuscript.

Regarding VP16-Stat3, it is important to recognize that its function in this experiment is purely as a positive control. Our experience with screens is that each individual plate must contain a positive control, meaning cells that receive a plasmid that reliably produces the phenotype for which one is screening. Vp16-Stat3 was selected to play this role based on its proven ability to boost neurite length above the level of Klf6 alone, the trait we sought to detect in candidate TFs. The purpose of the present screen was to probe the activity of naturally occurring TFs, and accordingly Vp16 domains were not part of the experimental design. In the future it may be interesting to test whether stronger phenotypes might result from a vp16 or -64 based modification, but we consider this experiment outside the scope of the present work. We have added language to the results section to clarify the role of VP16-Stat in the screening experiment, and our use of non-VP16 constructs in the screen.

4. The experimental design in Figure 4 is also unclear and confusing. Nuclei are sorted for RNAseq analysis to probe the underlying mechanisms of growth promotion. According to the diagram the injured neurons in Figure 4a are collected. But in all other experiments, the effect of klf6 and synergizing TF is assessed on the non-injured sprouting neurons. Does the injury in itself change gene expression in CSMN? How does the analysis of gene expression in injured CSMN relate to the observed sprouting effect from non-injured neurons? If Klf6 combined with Nr5a2 were shown to also enhance re-growth of damaged neurons (see point #1 above), then this RNAseq experiment would be more relevant.

We apologize for the confusion, which stems from a left/right error during construction of the figure. To be clear, the nuclei collected for the RNAseq experiment, and now for an additional single-nuclei experiment, were gathered entirely from the uninjured half of the cortex, not from nuclei whose axons were cut by the pyramidotomy. The directly injured nuclei are certainly of interest, but are not part of the present data. The figure has been updated to reflect the unilateral collection of only spared CST neurons, and the revised Methods also emphasizes this point.

5. The similarity in gene expression changes provided by klf6 expression in vitro (from their previous study Wang et al 2018) and the current in vivo data set is very interesting and warrants more discussion. This similarity highlights the usefulness of in vitro approaches.

We thank the reviewer for pointing this out. We have expanded the discussion to include specifics on common gene modules that are evoked in response to Klf6 overexpression *in vitro* and *in vivo*.

6. A clear evidence for co-expression of the two transcription factors in the same neuron together with the tracer is necessary, especially because this is the main claim of this study.

The presented in situ images in Supp figure 4 are not convincing as presented. It is not clear in which cell type the signal is detected, as there is no co-staining with a neuronal marker. Quantification of the percentage of cells expressing both transcripts should be shown. How were the RNA scope probe designed to detect specifically the expressed but not endogenous TF?

This is an important point, and we have addressed it with two additional experiments. The first was a general test of the viral co-injection strategy, and also speaks to the issue of cell type. In adult mice, AAVs carrying fluorescent reporters were injected to the cortex while CST neurons were labeled by retrograde tracing. Two weeks later we examined the rate of co-transduction (i.e. co-detection of both fluorophores) specifically in CST neurons. In five replicate animals, with more than 200 CST neurons scored in each, in >95% of cases expression of one fluorophore was associated with expression of the second. These data support the premise that a mixed-AAV strategy is largely effective in co-expressing two transgenes within the target population of CST neurons. As discussed in a point below, this test was performed with AAV2-Retro, in order to validate the use of this serotype in an “anterograde” fashion.

The second experiment directly examined co-expression of *Klf6* and *Nr5a2*, along with tdTomato tracer. These cortical injections matched the viral ratios used in the axon growth studies. We examined co-expression of *Klf6*, and *Nr5a2* using RNAscope. The use of fresh-frozen tissue and the RNAscope procedure itself acted to diminish the strength and quality of the tdTomato signal, but at high magnification it remained possible to identify individual cells that expressed tdTomato. We quantified 475 cells from three replicate animals, and found that >90% of tdTomato+ cells also co-expressed *Klf6* and *Nr5a2*. We have added these data as **Supplementary Figure 6**, along with representative images that we hope the reviewer will find more convincing than the originals.

Regarding the question of probe specificity, note that our probes are not designed to distinguish between endogenous and virally-expressed transgenes. Owing to the developmental downregulation of *Klf6* and *Nr5a2*, endogenous expression is low in cortical tissue. Thus the strongly elevated expression specifically in the area of injection, illustrated in **Supp Fig 6**, indicates effective viral overexpression.

Minor points

• Supp figure 1b should be presented in main figure, as this is important to understand what is measured.

As requested, the Supp Figure 1b is now present as **Figure 2d** in the main text.

• There are no references in the text for fig.2b,c

These have been added (note they are now 3b and 3c in the revised Figures).

• Figure2- h and l both have images describing co-expression of *Klf6+Nkx3.2*, but the effects are different

The labeling error has been corrected, thank you.

- **Venn diagrams of DE genes in the RNAseq related to Figure 4 would be helpful to understand the differences between single and double TF expression.**

We have now added a Venn diagram to figure 7 (original figure 4), to illustrate overlap and differences in transcripts following single and double TF treatments. We thank the reviewers for this suggestion.

- **Table Supplementary 3 - There are some mix-ups in the virus descriptions.**

We have fixed the transposed gene names in Supplementary Table 3, thank you.

- **Add explanation why for the TF expression AAV2 was used and for the TdTomato expression AAV9 was used - is there a difference in infection efficiency or cell specificity of expression?**

We used AAV2-Retro to deliver transgenes to CST neurons; this information was conveyed in the original Supplementary Table 3 but not properly emphasized in the text. The selection of Retro-AAV reflects a larger strategic decision by the lab to invest in retrograde vectors for multiple projects, and the economic reality that it would be difficult to acquire multiple serotypes of all targets in large combinatorial experiments (i.e. we had to choose one, and went with the versatile version for future directions). To be clear, in the injury experiments we do not specifically use the retrograde properties of AAV2-Retro, but instead take advantage of our findings that it can be used very successfully in an “anterograde” fashion as well, meaning strong transduction when applied to cell bodies. In new Supplementary Figure 2 we validate this use of AAV2-Retro, injected directly to the cortex, for local CST transduction. Axon tracers, however, were AAV9 and not AAV2-Retro, based on our concern at the time (now diminished) that retrograde spread of fluorophores could complicate tracing in the spinal cord. Critically, effective co-expression of tdTomato, Klf6 and Nr5a2 with this mixture is now quantified in **Supplementary Fig. 6**. We apologize again for the confusion and hope that new Supp Figs. 2 and 6, along with the new text added to the results sections, clears it up.

- **Overall a better definition is needed when using the terms axonal growth, extension, sprouting and regeneration.**

We recognize the terminology is fraught, and in the revision follow the general framework of Tuszynski and Steward in their 2012 Neuron “Concepts” Primer. We avoid using the term regeneration, which can be taken to imply growth that extends beyond an injury site; this is not among the findings we report here. We are now similarly careful with the term “sprouting,” again because some would interpret this to imply growth specifically from directly injured axons. Accordingly, the term sprouting is used here only in reference to the new experiment involving directly axotomized CST neurons. Overall we favor the term “growth,” as it agnostic with

respect to mechanism and injury status, and denotes the production of new axonal material that is spurred by Klf6/Nr5a2 treatment. We appreciate the good advice to take care with the terms.

• **There are no details on FANS in the method section.**

We have updated the methods section to include details on FANS, thank you.

• **Figure 4g-I Regulatory network analysis of genes upregulated after combined Klf6 and Nr5a2. Is that all the genes regulated by klf6+Nr5a2 or only the 192 that were not change by either TF alone.**

The network analysis was only done on the 192 genes that are uniquely regulated by klf6+Nr5a2 (not by either TF alone). We have re-worded the legend to clarify this and the addition of a Venn diagram further clarifies this.

Reviewer #3 (Remarks to the Author):

In their manuscript, Venkatesh and colleagues present a pipeline for the identification of transcription factors (TFs) that could promote axon growth and regeneration, especially when provided together. To identify such potential synergy, the authors use Klf6, known to promote neurite growth following ectopic expression, as a starting point. First, computational methods are used to identify other TFs that co-occupy regulatory DNA with Klf6. This is followed by a high-throughput screen to explore which of these TFs actually promotes axon growth from cortical neurons when co-transfected with Klf6. The outcome of this screen is subsequently used to perform network analyses that identify the ‘core’ TFs Nr5a2, Eomes and Rarb. AAV-mediated overexpression of these TFs in the cortex following CST injury is then induced and shown to facilitate enhanced regeneration, especially when combinations of TFs are provided. Finally, neurons are collected from the transduced cortex and subjected

to RNAseq to begin to provide insight into underlying molecular mechanisms. In all, this is a nice study that emphasis the strength of combing TFs for enhancing axon (re)growth and that provides a pipeline for the identification of the most powerful candidates.

I have a few points that need to be addressed prior publication:

- The in vitro screen presented in Fig. 1b lacks morphological details. No transfected neurons are shown and it is hard to assess the quality/robustness of this screen. Are there other effects, e.g. on cell death, axon branching etc. The presentation of these data needs to

be expanded, perhaps as part of the Suppl. Data. Are there any quality controls for the screen?

As suggested, we have added a new **Supplementary figure 1** that provides expanded information regarding morphological effects in the screen (e.g. length of the longest neurite, number of neurites, and length of neurites with and without branches included). We have also expanded our description of the screen in the relevant Results section, with a focus on steps taken for quality control, and included example images in the main text. In practical terms, a central quality control measure is plate layouts that include positive and negative controls on each plate; this system is now better described in Results and Methods. Note that our screen did not have cell survival as a direct readout, although the consistency of cell numbers across plates can act as indirect indicator that large cell death effects were not present. In addition we emphasize that for Klf6/Nr5a2 we have added new data to test for cell survival effects *in vivo* (see below).

- In line 157-164, data are shown that identify TFs related to Klf6 (by analysis of neurons overexpressing Klf6) that are also found in the previous computational approach. It makes me wonder which is the added benefit of the analysis as shown in Fig. 1. Wouldn't the identification of TFs following overexpression of a specific TF (such as Klf6) be a more straightforward approach?

We appreciate the reviewer's drive to maximize efficiency and streamline the discovery process. We would emphasize, however, the critical need for multiple approaches to achieve triangulation. Our experience is that any single bioinformatic framework has the potential for error, especially false positives. The overexpression analysis considers gene changes evoked by Klf6 (in cultured neurons, without distinguishing between direct and indirect targets), and uses only motif analysis specifically in promoter regions to predict cooperating TFs. The pipeline we developed in Figure 1 starts from developmentally regulated genes, considers both promoter and enhancer regions, and deploys much more sophisticated methods to predict TF binding. Each pipeline has strengths and weaknesses, and depending on stringency settings could each potentially predict a wide range of TFs. We consider the existence of shared target TFs between the two pipelines a strength of our integrated approach, and would hesitate to label either pipeline as dispensable.

- The focus of this manuscript is very much on axon growth but the in vivo effects presented could also be a result (in part) of enhanced branching, increased survival, reduced sensitivity to an inhibitory environment etc. At least some experiments need to be performed to control for this.

We agree that the manuscript would benefit from consideration of alternate explanations, and have now performed two additional experiments as well as reimaging and analyzing existing tissue.

Regarding cell survival, in **Supplemental Figure 7** we show results from new experiments that tested Klf6/Nr5a2-mediated effects on cell death in the days after injection, as well as a test for

longer term survival of identified CST neurons in the vicinity of the Klf6/Nr5a2 injection. At both time points, levels of cell death are low and indistinguishable between control and Klf6/Nr5a2-treated cortex.

The suggestion to examine effects on branching was excellent, and led us to uncover an additional effect of Klf6/Nr5a2 expression. We re-imaged CST axons in cervical spinal as they sprouted across the midline in both control and Klf6/Nr5a2-treated animals, and manually traced axons, selecting those that ran in the plane of the image for a minimum of 100 μ m. We then counted the number of branch points, normalized to the traced length, and found that Klf6/Nr5a2 produced a reliable and striking increase in the frequency of branch points. These data have been added to a new **Figure 6**. We thank the reviewer for this suggestion, which has enriched our understanding of Klf6/Nr5a2's phenotypic effects.

- Fig. 3e: Why is there no effect of NR5A2+RARB at 200 micron while there is at 400 and 600 micron?

We can only speculate. Because CST axons do not necessarily travel in the plane of the transverse tissue section, they can “appear” at any distance from the midline without being visible in the intervening space. In Fig. 3e one can see a trend toward elevated growth at the 200 μ m distance: seven Nr5a2+Rarb animals fell above the 3rd quartile of the comparison Rarb-only group, eight above the average – yet the remaining two showed very low counts at the 200 μ m distance. Thus it may simply be an issue of variability as axons passed in and out of the plane of the tissue sections in an “unlucky” way in two animals. Alternatively, from a morphological perspective, counts of axons intersections can sometimes be lower near the midline, because CST may extend for some distance into contralateral grey matter before initiating branching. We emphasize that our practice of gathering data at multiple distances from the midline, and from 4 evenly spaced sections, is designed to detect phenotypic effects in the face of such sources of variability listed above.

- Fig. 4: This part is very open-ended. At least some of the regulated genes should be confirmed in vivo and for this type of journal one wonders whether functionally counteracting a downstream gene in vitro or in vivo following TF overexpression is not a required minimum.

We agree with the importance of supporting these data with an independent approach, and considered carefully the best way to do so. A main conclusion from the RNAseq data is that the regulated gene effects are “broad but shallow”, meaning that 100s of genes change in expression but almost no targets show an all or none response. Moreover, CST neurons comprise just a few percent of the cortex and there is no guarantee that Klf6/Nr5a2 treatment would behave similarly in other cortical cell types that do not project to the site of injury. We considered a strategy of spot-checking targets with a qPCR or RNAscope approach, but given the modest size of the predicted change for any single target and the relative scarcity of affected cells, were not confident this approach would yield strong validation.

Accordingly, we proceeded with an alternate single-cell based experiment. The key advantages are the ability to look broadly at the set of predicted target genes in definitively identified CST neurons in which either control or both Klf6 and Nr5a2 are expressed, while employing a separate and independent method of RNA detection, sequencing and bioinformatics analyses. The results of this validation experiment are presented in **Supplementary Fig. 8**, and provide independent confirmation of the key findings of the original RNAseq analyses (now shown in revised Figure 7). 144 target genes (>50% overlap) were again found to increase in expression upon dual expression of Klf6 and Nr5a2 (now confirmed by direct detection of these transcripts within the analyzed nuclei), which critically are comprised almost entirely of transcripts that fall within functional clusters of Macromolecule Biosynthesis, DNA repair and CNS development. Overall, this new analysis strengthens the core conclusion that forced expression of Klf6 and Nr5a2 increases the expression of transcripts involved in Macromolecule synthesis, CNS Development, and DNA repair.

Finally, we understand the reviewer's comment about possibly proceeding with functional tests of target genes, but we would emphasize again the "broad but shallow" nature of the changes in gene expression. Our interpretation is that Klf6 and Nr5a2 exert effects through coordinated yet modest changes in large gene networks, not through single effector genes. In this conception, it would likely be quite difficult to detect changes in axon growth by knocking out or overexpressing single targets. We hope the reviewer agrees that even in the absence of identifying a single effector gene, the creation of a novel bioinformatic pipeline, the discovery of novel TF synergies for axon growth, profiling transcriptional consequences across two independent platforms, and the now-expanded characterization of the *in vivo* phenotypes from lead TFs stand as sufficient advance.

Reviewers' comments:

Reviewer #1 (Remarks to the Author):

The authors have carefully evaluated the concerns raised by this and other reviewers. New experiments were carried out to either clarify or further support their conclusions. No further comments.

Reviewer #2 (Remarks to the Author):

The authors have very nicely answered my comments as well as the comments from other reviewers. Specifically, the authors have now tested if the combined TF strategy can promote regeneration of cut axons (new figure 6). They can now conclude that Klf6/Nr5a2 expression is insufficient to confer regenerative ability in the inhibitory lesion environment, but Klf6/Nr5a2 expression does act on injured neurons to increase collateral extension. This is an important result for the field. The authors also provided a very nice validation by RNAscope of the co-expression of KLF6, Nr5a2 and the tracer (Supp fig 6). The authors should check the updated publication status when citing Biorxiv papers. Overall, this manuscript has been greatly improved with additional data, better data presentation and clarifications in the text and I fully support publication.

Reviewer #3 (Remarks to the Author):

The authors have extensively revised their manuscript and addressed all of my concerns. They added a series of impressive data making the manuscript even stronger and interesting. This is an excellent study and I fully support publication.

Reviewer #1 (Remarks to the Author):

The authors have carefully evaluated the concerns raised by this and other reviewers. New experiments were carried out to either clarify or further support their conclusions. No further comments.

We thank the reviewer for their valuable comments which has helped strengthen the findings in this manuscript.

Reviewer #2 (Remarks to the Author):

The authors have very nicely answered my comments as well as the comments from other reviewers. Specifically, the authors have now tested if the combined TF strategy can promote regeneration of cut axons (new figure 6). They can now conclude that Klf6/Nr5a2 expression is insufficient to confer regenerative ability in the inhibitory lesion environment, but Klf6/Nr5a2 expression does act on injured neurons to increase collateral extension. This is an important result for the field. The authors also provided a very nice validation by RNAscope of the co-expression of KLF6, Nr5a2 and the tracer (Supp fig 6). The authors should check the updated publication status when citing Biorxiv papers. Overall, this manuscript has been greatly improved with additional data, better data presentation and clarifications in the text and I fully support publication.

We appreciate the positive comments. We have now checked all preprint citations and updated them to the published versions.

Reviewer #3 (Remarks to the Author):

The authors have extensively revised their manuscript and addressed all of my concerns. They added a series of impressive data making the manuscript even stronger and interesting. This is an excellent study and I fully support publication.

We thank the reviewer again for input and guidance to improve the manuscript.